# Identifying multi-compartment Hodgkin-Huxley models with high-density extracellular voltage recordings

**Ian Christopher Tanoh**[1,2]    **Michael Deistler**[3,4]    **Jakob H. Macke**[3,4,5]
**Scott W. Linderman**[1,2]
[1]Department of Statistics, Stanford University
[2]Wu Tsai Neurosciences Institute, Stanford University
[3]Machine Learning in Science, University of Tübingen
[4]Tübingen AI Center, Tübingen
[5]Department of Empirical Inference, Max Planck Institute for Intelligent Systems, Tübingen
{ictanoh, scott.linderman}@stanford.edu

## Abstract

Multi-compartment Hodgkin-Huxley models are biophysical models of how electrical signals propagate throughout a neuron, and they form the basis of our knowledge of neural computation at the cellular level. However, these models have many free parameters that must be estimated for each cell, and existing fitting methods rely on intracellular voltage measurements that are highly challenging to obtain *in vivo*. Recent advances in neural recording technology with high-density probes and arrays enable dense sampling of extracellular voltage from many sites surrounding a neuron, allowing indirect measurement of many compartments of a cell simultaneously. Here, we propose a method for inferring the underlying membrane voltage, biophysical parameters, and the neuron's position relative to the probe, using extracellular measurements alone. We use an Extended Kalman Filter to infer membrane voltage and channel states using efficient, differentiable simulators. Then, we learn the model parameters by maximizing the marginal likelihood using gradient-based methods. We demonstrate the performance of this approach using simulated data and real neuron morphologies.

## 1 Introduction

Biological neurons are equipped with a wide range of biophysical mechanisms that contribute to information processing [1–3] and to the brain's robustness to environmental perturbations [4, 5]. Multi-compartment Hodgkin-Huxley models provide a principled framework for studying biophysical mechanisms of neurons [6, 7]. Multi-compartment models treat neurons as connected electrical compartments, allowing simulation of how voltage signals propagate throughout a cell as ion channels open and close. A central limitation for building multi-compartment models—and for understanding biophysical mechanisms in neurons—is the lack of experimental techniques to precisely measure the biophysical parameters of these models. As such, these parameters must be inferred from recordings of the activity of neurons [8–11]. Typically, these recordings are made from within a neuron using intracellular measurements such as patch-clamp electrophysiology, which enables precise recordings of voltage dynamics in the soma [12, 13] and, for large neurons, in dendrites [14]. However, intracellular methods are technically challenging, and only provide voltage signals from a very small number of locations within a neuron (typically, only from the soma) [15].

39th Conference on Neural Information Processing Systems (NeurIPS 2025).

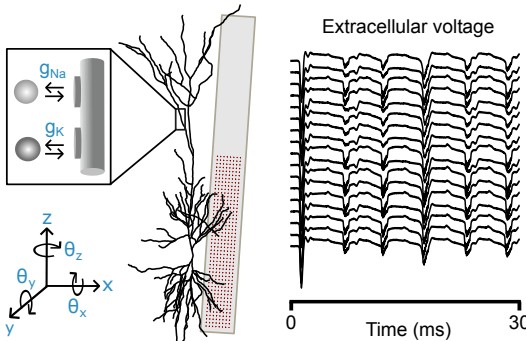

Figure 1: **Our goal: Inferring biophysical properties of single neurons from dense extracellular measurements:** We want to identify key biophysical properties—maximum membrane conductances of ion channels and the cell's relative position to the probe (left, blue) from extracellular voltage traces recorded from multiple sites along the probe (right), capturing the neuron's activity over time.

Recent advances in neural recording technologies have opened new possibilities for studying neuron biophysics *in vivo*. High-density silicon probes such as the Neuropixel Ultra [16] can record extracellular voltage traces across hundreds of closely spaced electrodes. These signals, which reflect the transmembrane currents generated by nearby neurons, are traditionally used for spike detection and sorting—identifying when and where neurons fire spikes [17, 18].

While effective for tracking population activity, this approach treats spikes as discrete events and ignores the exquisite biophysical details of neurons [19].

In this work, we revisit the extracellular signal, not just as a source of discrete spike times, but as a rich, spatially structured observation from which we can infer biophysical properties of individual neurons. This is made possible by three key developments: (1) high-density probes and 2D arrays that allow us to sample extracellular voltage at many sites around a cell to triangulate its position and morphology; (2) differentiable simulators like JAXLEY [20] that enable fast, hardware-accelerated simulation of complex biophysical models and gradient-based parameter optimization; and (3) scalable state-space inference frameworks such as DYNAMAX [21], which allow efficient state inference and parameter estimation in high-dimensional, nonlinear dynamical systems.

We leverage these developments to provide a statistical approach for inferring the parameters of multi-compartment biophysical models from extracellular observations: We extend classical multi-compartment Hodgkin–Huxley models to include a biophysical model of extracellular potentials. We then use an Extended Kalman Filter (EKF) [22] to infer latent neuronal states—such as membrane voltages and gating variables—from observed extracellular signals and we improve the computational efficiency with diagonal and block-diagonal approximations to the filtering covariance. To estimate model parameters, including ion channel densities and the neuron's spatial position relative to the probe, we maximize the marginal likelihood using gradient-based optimization.

We first demonstrate that our method can successfully infer membrane properties and cell locations in simplified neuron models, and that the use of an EKF largely improves the method's robustness to model misspecification. We then apply our method to synthetic extracellular data generated from real neuron morphologies and demonstrate that the method can scale to morphologies with more than a hundred compartments. Overall, our results suggest that the combination of differentiable simulation and state-space modeling makes it possible to estimate biophysical properties across the entire morphology of a neuron based only on extracellular voltage recordings.

## 2 Biophysical Modeling of Neurons

### 2.1 Modeling Action Potentials with Hodgkin–Huxley Dynamics

The multi-compartment Hodgkin–Huxley model (HH) describes how ion channels shape the membrane potential across branches of a neuron. Each compartment represents a small component of the cell with its own voltage, capacitance, and set of ion channels. Let $V^{(i)}$ denote the membrane

potential of compartment $i$, with surface area $S^{(i)}$ and specific membrane capacitance $C_m$. Each compartment has a set of ion channels, $\mathcal{C}$, and each channel $c \in \mathcal{C}$ is characterized by a maximum conductance $g_c^{(i)}$, reversal potential $E_c$, and a time-varying state $\lambda_c^{(i)}$. A nonlinear activation function, $a_c(\lambda_c^{(i)}) \in [0, 1]$, specifies the fraction of the maximum current that can flow through the channel, as a function of its state. The voltage dynamics for compartment $i$ are given by,

$$C_m \frac{\mathrm{d}V^{(i)}}{\mathrm{d}t} = \frac{I_{\text{ext}}^{(i)}}{S^{(i)}} - \sum_{c \in \mathcal{C}} g_c^{(i)} a_c(\lambda_c^{(i)})(V^{(i)} - E_c) + \sum_{j \sim i} g^{(i,j)}(V^{(j)} - V^{(i)}), \tag{1}$$

where $I_{\text{ext}}^{(i)}$ is the external current applied to compartment $i$, and $g^{(i,j)}$ is the axial conductance coupling compartment $i$ to its neighbor $j$. The total transmembrane current in compartment $i$ is denoted by $I_m^{(i)}$ and is the sum of the ionic current $I_{\text{ion}}^{(i)} = S^{(i)} \sum_{c \in \mathcal{C}} g_c^{(i)} a_c(\lambda_c^{(i)})(V^{(i)} - E_c)$ and the capacitive current $I_{\text{cap}}^{(i)} = S^{(i)} C_m \frac{\mathrm{d}V^{(i)}}{\mathrm{d}t}$ [23]. For instance, in the original single compartment HH model with sodium, potassium, and leak channels, the total current is,

$$I_m = S \left( g_{\text{Na}} m^3 h(V - E_{\text{Na}}) + g_{\text{K}} n^4 (V - E_{\text{K}}) + g_{\text{L}}(V - E_{\text{L}}) + C_m \frac{\mathrm{d}V}{\mathrm{d}t} \right). \tag{2}$$

Here, for example, the states of the sodium channel are $\lambda_{\text{Na}} = (m, h) \in [0, 1]^2$ and the activation function is $a_{\text{Na}}(\lambda_{\text{Na}}) = m^3 h$. The gating variables evolve according to first-order kinetics $\frac{\mathrm{d}\lambda_c}{\mathrm{d}t} = \alpha_c(V)(1 - \lambda_c) - \beta_c(V)\lambda_c$, where $\alpha_c(V)$ and $\beta_c(V)$ are voltage-dependent rate functions that control the opening and closing transitions of the ion channel. Specific forms for the $\alpha$ and $\beta$ functions are available in Dayan and Abbott [24], e.g. Importantly, note that by eq. (1), the ionic current and the capacitive current cancel out to give the following expression of the transmembrane current:

$$I_m^{(i)} = I_{\text{ext}}^{(i)} + S^{(i)} \sum_{j \sim i} g^{(i,j)}(V^{(j)} - V^{(i)}). \tag{3}$$

## 2.2 Modeling Extracellular Voltage Measurements

Extracellular voltages reflect the collective activity of nearby neurons. While the physics of extracellular recordings are well understood [25, 26], our work is, to our knowledge, the first to leverage this framework for fitting multi-compartment HH models to extracellular measurements. Let $\Phi$ denote the extracellular potential measured by a fixed site on a recording probe. We treat each compartment as a point source of transmembrane current located at its center and denote $r_i$ the distance from the recording site to compartment $i$ of a neuron. It follows from Coulomb's law that the measured extracellular voltage is,

$$\Phi \propto \sum_{i=1}^{N} \frac{I_m^{(i)}}{r_i}. \tag{4}$$

In words, the observed potential $\Phi$ is proportional to the sum of transmembrane currents from nearby compartments, weighted by the inverse of the distance to the compartment. Importantly, it follows from eq. (3) and (4) that the extracellular voltages at time $t$ are simply linear functions of the membrane voltages at time $t$.

## 3 State Inference and Parameter Estimation

We aim to infer the membrane potential and channel states and to estimate the biophysical and structural parameters of a neuron from noisy extracellular voltage traces. We can achieve both goals by casting the multi-compartment HH model as a state space model (SSM) with nonlinear dynamics. However, the nature and scale of these models present unique challenges, which we address below.

### 3.1 State Space Model Formulation

A state space model (SSM) is a latent variable model with latent states $\mathbf{z}_{1:T}$, observations $\mathbf{y}_{1:T}$, and parameters $\theta$. In a Markovian SSM, latent states evolve according to a dynamics distribution,

$p(\mathbf{z}_t \mid \mathbf{z}_{t-1}; \theta)$, and they give rise to observations via an emission distribution $p(\mathbf{y}_t \mid \mathbf{z}_t; \theta)$. There are a host of well-established techniques for state inference and parameter estimation in such models [22, 21], depending on the nature of the dynamics and emission distributions.

In our case, the model parameters, $\theta$, may include the maximum conductances of various ion channels as well as parameters governing the neuron's spatial configuration relative to the recording probe. The latent states consist of the voltages and channel states for each compartment, $\mathbf{z}_t = [V^{(1)}, \ldots, V^{(N)}, \{\lambda_c^{(1)}\}_{c \in \mathcal{C}_1}, \ldots, \{\lambda_c^{(N)}\}_{c \in \mathcal{C}_N}]$. To model the state dynamics, we use an Euler discretization of the multi-compartment HH differential equations (see eq. (1)),

$$\mathbf{z}_{t+1} \sim \mathcal{N}(f_\theta(\mathbf{z}_t, t), \Delta t \, \boldsymbol{\Sigma}_{\text{dyn}}), \tag{5}$$

where $f_\theta(\mathbf{z}_t, t)$ is the deterministic update function derived from the ODEs above, and $\boldsymbol{\Sigma}_{\text{dyn}}$ is a diagonal covariance matrix with entries from $\{\sigma_v^2, \sigma_{\text{state}}^2\}$. This Gaussian noise is intended to account for model uncertainty, such as missing ion channels or misspecified channel dynamics, which we will consider in the results section.

We assume that the only source of stochasticity in the emission process is measurement noise. Let $\mathbf{y}_t \in \mathbb{R}^M$ denote the observed extracellular potential at time $t$ across $M$ recording sites. It is modeled as,

$$\mathbf{y}_t \sim \mathcal{N}(h_\theta(\mathbf{z}_t), \sigma_{\text{obs}}^2 \mathbf{I}), \tag{6}$$

where $h_\theta(\mathbf{z}_t)$ is the predicted extracellular voltage from eq. (4) and $\sigma_{\text{obs}}^2$ is the observation noise variance. Since $h_\theta(\mathbf{z}_t)$ only depends on $\mathbf{z}_t$, the emission model is conditionally independent of previous states given the current latent state.

## 3.2 Scaling the Extended Kalman Filter to Large Biophysical Models

Since the dynamics are nonlinear, there is no closed-form solution for the filtering distribution $p_\theta(\mathbf{z}_{1:t} \mid \mathbf{y}_{1:t})$. To address this challenge, we use the extended Kalman filter (EKF), which linearizes the dynamics and emission functions via first-order Taylor expansions [22]. This linear approximation enables recursive estimation of the latent states, and it also provides an estimate of the marginal likelihood $p_\theta(\mathbf{y}_{1:T})$ for parameter estimation.

Biologically realistic neuron models often contain dozens or even hundreds of compartments, leading to latent state dimensions in the hundreds or thousands. The standard EKF maintains an estimate of the full state covariance matrix, and updating the filtering distribution requires solving a linear system with this matrix. Since this computation scales cubically with the number of states, it becomes computationally infeasible to apply standard EKF formulations to full-scale neuronal models. To overcome this challenge, we develop two scalable EKF variants for biophysical neuron model fitting.

**Diagonal approximation** Our first scalable variant adopts a diagonal approximation of the posterior covariance, inspired by Chang et al. [27], while exploiting the sparsity inherent in multi-compartment HH dynamics. Specifically, we approximate the EKF filtering distribution $p_t(\mathbf{z}) = \mathcal{N}(\mathbf{z}; \boldsymbol{\mu}_t, \boldsymbol{\Sigma}_{t|t})$ with a diagonal Gaussian $q_t(\mathbf{z}) = \mathcal{N}(\mathbf{z}; \tilde{\boldsymbol{\mu}}_t, \tilde{\boldsymbol{\Sigma}}_{t|t})$ where $\tilde{\boldsymbol{\Sigma}}_{t|t}$ and $\tilde{\boldsymbol{\mu}}_t$ are chosen to minimize the divergence $\text{KL}(q_t \| p_t)$. As shown by Chang et al. [27], this minimization admits a closed-form solution: the optimal diagonal Gaussian approximation $q_t$ matches the marginal means and precisions (inverse variances) of $p_t$. This leads to efficient recursive updates that preserve the EKF structure while requiring only diagonal storage—reducing memory cost from $\mathcal{O}(d^2)$ to $\mathcal{O}(d)$ for a $d$-dimensional state.

In practice, this approximation requires evaluating only the diagonal elements of two matrices: (i) $\mathbf{H}_t^\top \mathbf{R}_t^{-1} \mathbf{H}_t$, where $\mathbf{H}_t = \text{Jac}(h_\theta)(\boldsymbol{\mu}_{t|t-1})$ is the Jacobian of the emission model and $\mathbf{R}_t = \sigma_{\text{obs}}^2 \mathbf{I}$ is the observation noise covariance; and (ii) the predicted precision $\boldsymbol{\Sigma}_{t|t-1}^{-1}$, with

$$\boldsymbol{\Sigma}_{t|t-1} = \mathbf{F}_t \boldsymbol{\Sigma}_{t-1|t-1} \mathbf{F}_t^\top + \mathbf{Q}_t,$$

where $\mathbf{F}_t = \text{Jac}(f_\theta)(\boldsymbol{\mu}_{t|t-1})$ is the Jacobian of the dynamics and $\mathbf{Q}_t = \Delta t \, \boldsymbol{\Sigma}_{\text{dyn}}$ is the dynamics covariance (cf. Appendix B).

The diagonal of $\mathbf{H}_t^\top \mathbf{R}_t^{-1} \mathbf{H}_t$ can be computed in linear time with respect to the number of states [27]. Furthermore, because the measurement model maps the membrane voltages to extracellular voltages

through a fixed linear operator (eq. (3) and (4)), $\mathbf{H}_t^\top \mathbf{R}_t^{-1} \mathbf{H}_t$ is time-invariant, allowing its diagonal to be computed once and reused throughout filtering. The remaining challenge lies in efficiently computing $\mathrm{diag}(\boldsymbol{\Sigma}_{t|t-1}^{-1})$. To simplify calculations, we make the additional assumption that $\boldsymbol{\Sigma}_{t|t-1}$ is diagonal, reducing the problem to computing $\mathrm{diag}(\boldsymbol{\Sigma}_{t|t-1})$. Here, we exploit the sparsity pattern of $\mathbf{F}_t$, which follows directly from the local coupling structure of the HH equations. While voltage updates may depend on all other states due to the implicit Euler solver used in JAXLEY [20], gating variables typically depend only on their own previous value and the voltage of their corresponding compartment. This sparsity enables efficient computation of $\mathrm{diag}(\boldsymbol{\Sigma}_{t|t-1}^{-1})$. In practice, we compute $\mathbf{F}_t$ using the SPARSEJAC library [28], which constructs Jacobians directly from sparsity patterns without instantiating the full dense matrix.

When is the diagonal approximation reasonable? We assume that the initial filtered covariance matrix $\boldsymbol{\Sigma}_0$ is diagonal. The EKF covariance update is:

$$\boldsymbol{\Sigma}_{t|t} = \left( \boldsymbol{\Sigma}_{t|t-1}^{-1} + \mathbf{H}_t^\top \mathbf{R}_t^{-1} \mathbf{H}_t \right)^{-1}.$$

The procedure we described is equivalent to performing this update step with both $\boldsymbol{\Sigma}_{t|t-1}$ and $\mathbf{H}_t^\top \mathbf{R}_t^{-1} \mathbf{H}_t$ diagonal. Is this valid?

In Appendix B, we show that for well-behaved HH kinetics—smooth transition rates, small integration steps, and not too frequent spiking—the dynamics Jacobian exhibits limited cross-coupling for morphologies of moderate complexity. Under these conditions, a diagonal prior covariance and dynamics covariance can yield an approximately diagonal predicted covariance $\boldsymbol{\Sigma}_{t|t-1}$.

The tighter constraint arises from the emission model. The time-invariant matrix $\mathbf{H}_t^\top \mathbf{R}_t^{-1} \mathbf{H}_t$ reflects how strongly electrode signals overlap across compartments. When a neuron is discretized into many compartments and the electrodes are densely packed around it—as in Neuropixels or similar high-density probes—its off-diagonal entries can become large, and approximating it by a diagonal can severely degrade performance (cf. Appendix B for a complete derivation). Thus, the diagonal EKF appears best suited for electrode configurations in which each sensor predominantly captures activity from only a few nearby compartments.

**Block-diagonal approximation** The diagonal EKF procedure described above assumes $\mathbf{H}_t^\top \mathbf{R}_t^{-1} \mathbf{H}_t$ to be diagonal. When this assumption fails, the true update step no longer preserves diagonality even for well-behaved HH models; enforcing it by truncation discards key aspects of the geometry of the problem and harms accuracy.

To avoid this loss, we introduce a more structured approximation that remains consistent under EKF updates. Since the emission model depends only on voltages, $\mathbf{H}_t^\top \mathbf{R}_t^{-1} \mathbf{H}_t$ is block-diagonal, with a single nonzero block corresponding to the voltage dimensions. This motivates a *block-diagonal* covariance structure, where the EKF predicted and filtered covariances consist of a dense voltage block and diagonal blocks for all remaining (gating) states. Concretely, let the first $K$ dimensions of the latent state vector correspond to compartment voltages and the remaining ones to gating variables. At each time step $t$, we assume that both the predicted and filtered covariance matrices take the form

$$\boldsymbol{\Sigma}_{t|\star} = \begin{bmatrix} \boldsymbol{\Sigma}_{t|\star}^{vv} & 0 \\ 0 & \mathrm{diag}(\boldsymbol{\sigma}_{t|\star}^2) \end{bmatrix}, \qquad \star \in \{t-1, t\},$$

where $\boldsymbol{\Sigma}_{t|\star}^{vv} \in \mathbb{R}^{K \times K}$ denotes the dense voltage–voltage covariance block, and $\boldsymbol{\sigma}_{t|\star}^2$ collects the gating variances. We refer to this family of matrices as the *BD(K)* family (block-diagonal with $K$ compartments).

We can prove that if the dynamics Jacobian $\mathbf{F}_t$ is BD($K$), then the BD($K$) family is *closed* under the EKF prediction and update: if the initial covariance matrix $\boldsymbol{\Sigma}_0$ is BD($K$), then subsequent predicted and filtered covariances remain BD($K$) (cf. Proposition 1). For the same reasons as in the diagonal case, the BD($K$) approximation of covariance matrices can be effective for well-behaved HH models.

Computationally, the EKF update step with BD($K$) covariances reduces to (i) a $K \times K$ solve for the voltage block—since the measurement Jacobian $\mathbf{H}_t$ has support only on voltages—and (ii) element-wise variance updates for the diagonal gating blocks (cf. proof of Proposition 1). The prediction step similarly preserves structure by propagating $\boldsymbol{\Sigma}_{t-1|t-1}^{vv}$ through the voltage–voltage sub-Jacobian and

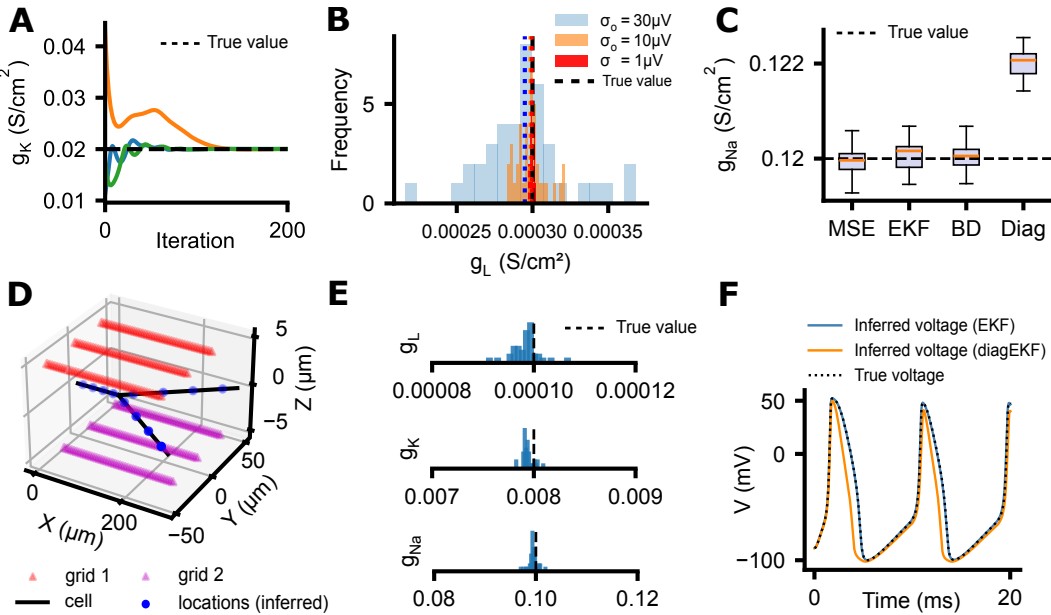

Figure 2: **Validation on simple morphologies.** Panels **A**–**C** show results for a single-branch cell. **A**: Estimated $g_K$ over training iterations for three random initializations (mean +/- 1 sd). **B**: Distribution of inferred $g_L$ under different observation noise levels $\sigma_{obs}$ (colored dashed lines indicate the mean). **C**: Distribution of inferred $g_{Na}$ for $\sigma_{obs} = 1\,\mu V$ for various inference methods (orange lines indicate the means). Panels **D**–**F** show results for a model with twelve compartments. **D**: A multi-branch neuron and two sets of voltage recording electrodes (red and purple). The EKF correctly identifies the true location of the neuron (EKF in blue, true locations as black lines). **E**: Conductances inferred across 40 datasets simulated with $\sigma_{obs} = 1\,\mu V$ via EKF for the 3rd branch of the cell. **F**: Recovery of membrane voltages for EKF and diagEKF in a compartment of the 3rd branch, averaged over the 40 aforementioned datasets.

updating gating variances via their local Jacobians, yielding overall complexity $\mathcal{O}(K^3 + d)$ instead of $\mathcal{O}(d^3)$ for a $d$-dimensional state (cf. proof of Proposition 1).

# 4 Validation on simple morphologies

## 4.1 Validation on single-branch neuron model

We first validated our method on a simple synthetic neuron: a 10-compartment branch with sodium and potassium HH channels (Appendix E.1). We initially assumed that the biophysical model was correctly specified and that the dynamics were fully deterministic. We stimulated the cell with constant current and recorded extracellular voltages from a single electrode placed $5\,\mu m$ from the soma. The goal was to infer maximum conductances assuming that the cell's position was known. We simulated 40 recordings corrupted by Gaussian noise ($\sigma = 1\,\mu V$) and initialized conductances within biologically plausible ranges. EKF-based inference accurately recovered the true parameters across random seeds, with negligible bias and variance (fig. 2A), supporting the validity of our approach.

Next, we assessed the effect of observation noise. Neuropixels Ultra probes exhibit RMS noise levels around $7.4\mu V$ in saline and approximately $9.4\mu V$ *in vivo* [16]. Moreover, in practice we expect voltage traces to be corrupted by currents from other nearby neurons, which our model treats as noise as well. Fixing the cell–probe distance at $5\mu m$, we simulated 40 noisy EAPs at three increasing noise levels, ranging from $1\mu V$ to $30\mu V$. As expected, we found that higher noise leads to higher variance in the parameter estimates (fig. 2B). Nevertheless, estimates remained very close to the ground truth even at $30\mu V$ RMS, which represents a challenging, high-noise regime (fig. E.3).

We also examined the effect of increasing the distance between the cell and the recording site. Since extracellular voltage scales proportionally with $1/r$, the signal-to-noise ratio declined with distance.

In line with this intuition, we found that larger distance between cell and recording site leads to higher variance in the estimate of membrane conductances (fig. E.1B).

Finally, we compared multiple inference approaches—standard EKF, diagonal EKF, block-diagonal EKF, and direct mean-squared-error (MSE) minimization as used in the original JAXLEY paper [20]. The MSE baseline replaces the probabilistic objective with deterministic dynamics and directly minimizes the squared error between observed and predicted extracellular potentials. Using the same setup (5 $\mu$m distance, 1 $\mu$V observation noise) and fitting the same 40 random traces, we found that the standard EKF, block-diagonal EKF, and MSE approaches achieved nearly identical estimates (fig. 2C). This confirms that the local linearization in the EKF introduces negligible bias and that the block-diagonal approximation provides an excellent surrogate for the full covariance in HH models. The diagonal EKF exhibited slightly higher bias but its voltage reconstructions remain highly accurate (fig. E.2). This is consistent with our analysis in Appendix B: only the first compartment lies near the electrode, and it has a single neighboring compartment, leading to minimal cross-visibility and making a diagonal covariance approximation efficient.

## 4.2 Validation on a multi-branch neuron model

We next evaluated our method on a more realistic multi-branch setting, assuming a correctly specified biophysical model and fully deterministic dynamics. Using JAXLEY, we simulated a synthetic neuron with three branches, each divided into four compartments (twelve in total), containing standard HH ion channels. The maximum conductances varied independently across branches, yielding nine biophysical parameters to infer. Details of the setup are provided in Appendix E.2. Unlike the single-branch case, we did not assume full knowledge of the cell's spatial configuration: while branch number, connectivity, lengths, and radii were known, the 3D coordinates of branch endpoints were treated as unknown, introducing 18 additional geometric parameters for a total of 27.

A central question is whether these spatial locations are identifiable from extracellular recordings. With electrodes confined to a single plane (e.g., a Neuropixel probe), mirror reflections of the neuron across the plane produce identical extracellular voltages (cf. Appendix D). To account for this unidentifiability, we simulated recordings from two separate probe positions, ensuring that the sets of recording sites did not lie in the same plane (fig. 2D, red and purple). We also ensured that the sites span the entire neuron to capture signal from all branches.

We fixed the observation noise to $1\mu$V and generated 40 sets of extracellular voltage recordings. To initialize parameters, we optimized from multiple random starting points on one dataset and retained the best solution (highest marginal log-likelihood), which we then reused across all runs. In this setting, the EKF consistently recovered the 3D compartment locations (fig. 2D, blue vs black), conductances (fig. 2E), and membrane voltages (fig. 2F), demonstrating the robustness and accuracy of biophysical and geometric inference in multi-compartment neuron models with unknown locations and membrane conductances. The diagonal EKF exhibited somewhat larger bias in the recovered conductances compared with the single-branch case (fig. E.6), but the voltage traces remained accurate up to small shifts (fig. 2F), and the compartment positions were inferred correctly (fig. E.8). In this multi-branch geometry, compartments are long enough that most electrodes are close to only a single compartment, and each compartment has very few neighbors, which together limit cross-compartment measurement interactions and explains why the diagonal approximation still yields informative reconstructions (cf. Appendix B).

## 4.3 Robustness to model misspecification

In experimental settings, the true underlying biophysical dynamics of a neuron are rarely known with certainty. To evaluate the robustness of our approach under such model misspecification, we conducted experiments where the ground truth and inference models differ in their ion channel composition and dynamics. The detailed experimental setup and additional figures can be found in Appendix E.3.

**Omitted Ion Channels**  In our first experiment, we generated synthetic data from a single-branch cell containing both classical HH channels and two additional conductances: an M-type potassium current and a calcium current. During inference, we deliberately used a simplified model containing only the HH channels, omitting the additional ones (fig. 3A).

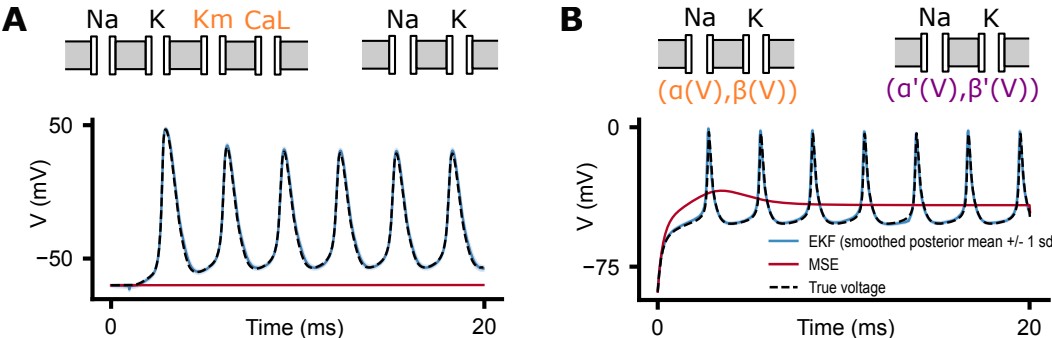

Figure 3: **Robustness to model misspecification.** Two types of channel misspecification are presented. **A**: missing channels (M-type $K^+$ and CaL). **B**: Incorrect gating dynamics for present channels. In both panels, recovered membrane voltage traces using MSE minimization and EKF are shown. EKF provides accurate voltage estimates despite misspecified channel models, whereas MSE minimization fails.

Since we allow for dynamics noise in our model with process noise, we were able to accurately recover the underlying membrane voltage, despite this substantial misspecification. In contrast, naive fitting via mean squared error (MSE) minimization failed to capture the dynamics, yielding voltage estimates that diverged substantially from the ground truth. This result highlights the importance of allowing for uncertainty in the state evolution, using our EKF-based approach.

**Perturbed Gating Dynamics**   We next evaluated the effect of incorrect gating kinetics. Here, we retained the correct channel types in the inference model but perturbed the rate constants $\alpha(V)$ and $\beta(V)$ used in gating variable dynamics, simulating errors in channel parameter specification (fig. 3B). Once again, EKF-based inference provided significantly more accurate estimates of both membrane voltage and channel states compared to MSE minimization. These results suggest that our approach can tolerate moderate misspecification in gating dynamics. Together, these experiments demonstrate that EKF-based inference, combined with stochastic modeling of dynamics, confers robustness to common forms of model misspecification.

## 5   Application to real morphologies

Having established the method's efficiency in controlled settings, we next apply it to morphologically detailed neuron models derived from real reconstructions, including a retinal ganglion cell (RGC) and a CA1 pyramidal neuron.

### 5.1   Application to 2D retinal ganglion cell

We applied our method to a morphologically detailed model of a retinal ganglion cell (RGC), replicating conditions typical of *in vitro* studies using planar multi-electrode arrays (MEAs) [29]. In these setups, retinal tissue is placed ganglion-side down on the MEA, enabling simultaneous current injection and extracellular voltage recording from hundreds of electrodes (fig. 4A). We adopted a reconstructed RGC morphology from Vilkhu et al. [29], consisting of a soma and a complex dendritic arbor discretized into 122 compartments. The membrane model includes modified HH sodium and potassium channels with non-standard kinetics. We assumed shared maximum conductances across all dendritic compartments, yielding a total of six parameters to estimate [29]. For details of the model, see Appendix F.

Here, we assumed the spatial location of the neuron to be known and constrained to a 2D plane, as is typical in retinal MEA experiments [30, 31, 29]. Extracellular voltages were recorded using a simulated MEA with $30\mu m$ pitch, matching the geometry of the original 512-electrode array. To reduce computation, we used a subset of 10 electrodes spanning the entire neuron.

Given the high dimensionality of the latent state space ($d = 610$), we used our scalable EKF variants for efficient inference. Notably, the diagonal EKF performed poorly: assuming diagonal covariances, the marginal log-likelihood diverged from its value at the true parameters, indicating severe model

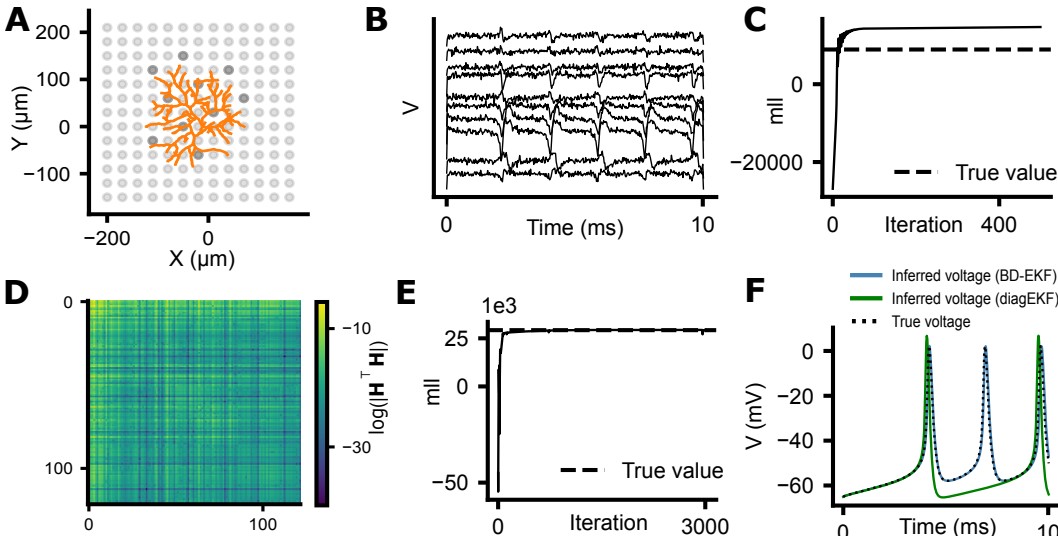

Figure 4: **Inference of retinal ganglion cell (RGC) properties given multi-electrode array (MEA) recordings.** **A**: RGC cell geometry and electrode layout. Darker shaded MEA sites match the voltage traces shown in panel B. **B**: Simulated extracellular recordings at ten sites for noise level $\sigma_{\mathrm{obs}} = 0.5\,\mu\mathrm{V}$. **C**: Marginal log-likelihood during optimization for diagonal EKF. **D**: The heatmap of the voltage block of $\mathbf{H}^\top\mathbf{H}$ (log scale) shows that the diagonal approximation is expected to fail. **E**: Marginal log-likelihood for block-diagonal EKF. **F**: Fitted voltage traces in the soma. The block-diagonal EKF infers the voltages accurately while the diagonal EKF fails.

mismatch (fig. 4C). The reconstructed voltage traces were also inaccurate, failing to reproduce the true dynamics (fig. 4F). This is precisely the failure mode predicted in Section 3.2: since each electrode records signal from many close compartments, $\mathbf{H}^\top\mathbf{H}$ cannot be diagonal (fig. 4D). In contrast, the block-diagonal EKF converged to the true marginal likelihood (fig. 4E) and accurately reconstructed both the membrane voltage traces (fig. 4F) and the underlying conductances (fig. F.1).

## 5.2 Application to CA1 hippocampal pyramidal cell

Finally, we applied our approach to a morphologically detailed model of a rat CA1 hippocampal pyramidal neuron, based on the C10 model introduced by Cutsuridis et al. [32]. This model has been used in prior computational studies to examine hippocampal circuit function and memory encoding, capturing the integration and propagation of signals across dendritic and axonal compartments [32, 33]. We recreated the full morphology of the C10 cell, including the soma, axon, basal dendrites, and apical dendrites (fig. 5A). The cell contained region-specific HH-type channels with subcellularly varying conductances and gating kinetics [32, 34, 35], yielding 24 maximum conductances to infer. Extracellular signals were recorded with a simulated Neuropixel Ultra probe replicating the real probe's geometry. To account for unknown spatial position, we added six rigid-body parameters (rotation and translation), for a total of 30 parameters. Further experimental details and figures are provided in Appendix G. Following Carnevale and Hines [7], the cell was discretized into 60 compartments, resulting in a latent state of dimension 360. We ran the block-diagonal EKF with over 100 restarts and retained the solution with the highest final marginal log-likelihood (fig. 5B).

Our method successfully recovers the neuron's spatial configuration (fig. 5A, black vs. blue), and accurately reconstructs membrane voltage (fig. 5D). It also recovers most of the maximum conductances, though not all (fig. 5C). This outcome is consistent with a well-documented phenomenon in conductance-based neuron models: parameter degeneracy, where multiple combinations of channel conductances can yield virtually indistinguishable voltage dynamics [36, 37]. Despite this, our method reliably recovers the neuron's functional dynamics and spatial structure, demonstrating that biophysical and geometric inference from high-density extracellular recordings is feasible in realistic, large-scale neuron models.

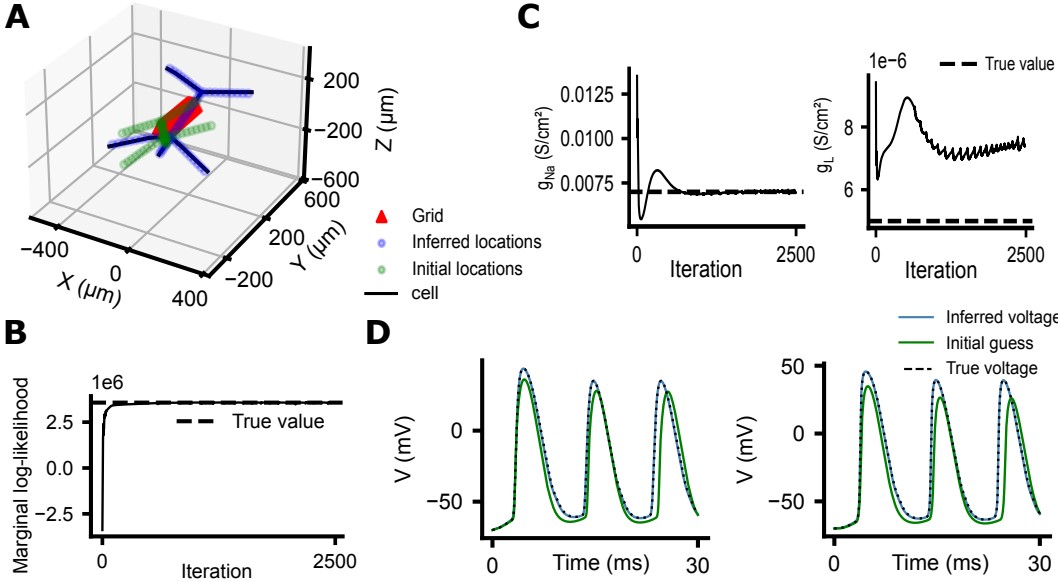

Figure 5: **Inference of membrane properties and locations of a C10 model of a pyramidal cell.** **A**: Pyramidal cell placed near a Neuropixel Ultra probe. Estimated compartment locations closely match true cell position. **B**: Marginal log-likelihood during optimization, converging to the ground truth value. **C**: Inferred conductances $g_{\mathrm{Na}}$ and $g_L$ in selected dendritic compartments. **D**: Recovered membrane voltage traces in dendrites of the stratum radiatum (left) and lacunosum-moleculare (right).

## 6    Conclusions, Limitations, and Future Work

**Summary**    We presented an approach for identifying detailed biophysical neuron models from high-density extracellular voltage recordings. Our method recovers not only membrane conductances but also the spatial position of a neuron relative to the recording probe, solely from extracellular data. We further developed a state-inference and parameter-estimation framework based on an Extended Kalman Filter (EKF), which provides robustness to model misspecification and accurate inference even when the assumed dynamics deviate from the true underlying system.

**Limitations and Future Work**    A limitation of the EKF for parameter estimation is its computational cost: even with diagonal or block-diagonal approximations, it remains more expensive than MSE minimization, particularly for large models. Additionally, EKF approximations may bias parameter recovery: while the block-diagonal EKF recovered parameters accurately in our evaluations, the diagonal approximation was less reliable in certain cell–probe geometries. Moreover, biophysical models often exhibit parameter degeneracy, where different conductance combinations yield similar voltage dynamics, limiting identifiability even when predictions fit the data well.

Finally, while our results demonstrate the feasibility of biophysical inference from extracellular recordings at the single-cell level in idealized settings, extending this framework to real *in vivo* conditions remains a significant challenge. Real recordings involve complex networks of interacting neurons. A natural next step is to move beyond isolated cells and model local neural populations surrounding the probe. Given knowledge of the targeted brain region, one could incorporate priors on network structure and cell types informed by anatomical and physiological studies, enabling more comprehensive modeling of *in vivo* extracellular signals.

**Conclusion**    Overall, our method—based on a combination of state-space modeling and differentiable simulation—opens up possibilities to estimate properties across the entire morphologies of neurons from extracellular signals alone. This approach will enable new analysis of how cells generate action potentials and open new avenues for studying the biophysical contributions to neural computation.

## Acknowledgments

We thank Alisa Levin, Noah Cowan, and other members of the Linderman Lab for helpful feedback throughout this project. We are also grateful to Amrith Lotlikar, Michael Sommeling, and E.J. Chichilnisky for sharing valuable insights into retinal ganglion cell physiology and model fitting approaches. This work was supported by grants from the NSF EFRI BRAID Program (2223827), the NIH BRAIN Initiative (U19NS113201, R01NS131987, R01NS113119, & RF1MH133778), the NSF/NIH CRCNS Program (R01NS130789), and the Sloan, Simons, and McKnight Foundations. It was also supported by the German Research Foundation (DFG) through Germany's Excellence Strategy (EXC 2064 – Project number 390727645) and the CRC 1233 "Robust Vision", and the European Union (ERC, "DeepCoMechTome", ref.101089288).

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

## A   Resources

All simulations and inference experiments were conducted using four open-source Python libraries:

- JAX (version 0.4.31, Apache License 2.0): A high-performance numerical computing library based on XLA, providing composable function transformations and GPU/TPU support. Available at: https://github.com/google/jax.
- JAXLEY [20] (version 0.7.0, Apache License 2.0): A differentiable simulator for multi-compartment Hodgkin–Huxley models, supporting GPU acceleration, just-in-time compilation, and gradient-based parameter optimization. Available at: https://github.com/jaxleyverse/jaxley.
- DYNAMAX [21] (version 0.1.4, MIT License): A library for state-space modeling and inference, used here for implementing the Extended Kalman Filter (EKF) and its (block) diagonal approximation. Available at: https://github.com/probml/dynamax.
- SPARSEJAC [28] (version 0.2.0, MIT License): A library for efficient sparse Jacobian computation using graph coloring and automatic differentiation. Available at: https://github.com/mfschubert/sparsejac.

For experiments on large-scale neuron models (e.g., the retinal ganglion cell and C10 model of a pyramidal neuron), we used an NVIDIA H100 GPU. All other experiments were run on a MacBook Pro with an Apple M1 Pro (8-core CPU, 16 GB RAM) running macOS 13.4.

All code used for inference, cell and voltage simulation, and the experiments presented in this paper is available at: https://github.com/ianctanoh/eap-fit-hh.

## B   Validity of the Diagonal EKF

The true update step of the EKF is:

$$\mathbf{\Sigma}_{t\,|\,t} = \left(\mathbf{\Sigma}_{t|t-1}^{-1} + \mathbf{H}_t^{\top}\mathbf{R}_t^{-1}\mathbf{H}_t\right)^{-1}. \tag{7}$$

The procedure described by Chang et al. [27] matches the diagonals of the precision matrices:

$$\mathrm{diag}(\mathbf{\Sigma}_{t|t}^{-1}) = \mathrm{diag}(\mathbf{\Sigma}_{t|t-1}^{-1}) + \mathrm{diag}(\mathbf{H}_t^{\top}\mathbf{R}_t^{-1}\mathbf{H}_t). \tag{8}$$

Since the filtered covariances are assumed to be diagonal, this amounts to:

$$\mathrm{diag}(\mathbf{\Sigma}_{t|t}) = \left(\mathrm{diag}(\mathbf{\Sigma}_{t|t-1}^{-1}) + \mathrm{diag}(\mathbf{H}_t^{\top}\mathbf{R}_t^{-1}\mathbf{H}_t)\right)^{-1}. \tag{9}$$

Their procedure is therefore equivalent to the standard EKF (7) with a diagonal predicted precision matrix $\mathbf{\Sigma}_{t|t-1}^{-1}$ and a diagonal measurement information matrix $\mathbf{H}_t^{\top}\mathbf{R}_t^{-1}\mathbf{H}_t$.

In their work, Chang et al. [27] assume that $\mathbf{F}_t = \mathbf{I}$. This drastically simplifies the computation of $\mathrm{diag}(\mathbf{\Sigma}_{t|t-1}^{-1})$ since $\mathbf{\Sigma}_{t|t-1} = \mathbf{\Sigma}_{t-1|t-1} + \mathbf{Q}_t$ and both $\mathbf{Q}_t$ and $\mathbf{\Sigma}_{t-1|t-1}$ are assumed to be diagonal.

This trivial Jacobian assumption is not realistic in our case. But to simplify calculations, we assume that $\mathbf{\Sigma}_{t|t-1}$ is diagonal, so that $\mathrm{diag}(\mathbf{\Sigma}_{t|t-1}^{-1}) = \mathrm{diag}(\mathbf{\Sigma}_{t|t-1})^{-1}$.

Our procedure becomes equivalent to the standard EKF with a diagonal predicted covariance matrix $\mathbf{\Sigma}_{t|t-1}$ and a diagonal measurement information matrix $\mathbf{H}_t^{\top}\mathbf{R}_t^{-1}\mathbf{H}_t$.

Hence, the validity of our diagonal EKF method relies jointly on the *dynamics model* (through $\mathbf{\Sigma}_{t|t-1}$) and the *emission model* (through $\mathbf{H}_t^{\top}\mathbf{R}_t^{-1}\mathbf{H}_t$). We discuss each in turn.

—

**A diagonal approximation of $\Sigma_{t|t-1}$ is reasonable for *well-behaved* HH models**

The predicted covariance verifies

$$\Sigma_{t|t-1} = \mathbf{F}_t \Sigma_{t-1|t-1} \mathbf{F}_t^\top + \mathbf{Q}_t, \tag{10}$$

where $\mathbf{F}_t = \mathrm{Jac}(f_\theta)(\boldsymbol{\mu}_{t-1|t-1})$ and $\mathbf{Q}_t$ is the dynamics covariance. Assume that both $\Sigma_{t-1|t-1}$ and $\mathbf{Q}_t$ are diagonal. The question becomes: What is the structure of $\mathbf{F}_t \Sigma_{t-1|t-1} \mathbf{F}_t^\top$?

***Structure of $\mathbf{F}_t$.*** In multi-compartment Hodgkin–Huxley models, each compartment voltage depends on its own state and on voltages of neighboring compartments, while each gating variable depends only on its own previous value and the voltage of its corresponding compartment. Ordering the latent state as

$$\mathbf{z}_t = \left[ \mathbf{v}_t, \, \mathbf{g}_t^{(1)}, \, \mathbf{g}_t^{(2)}, \, \ldots, \, \mathbf{g}_t^{(L)} \right],$$

the Jacobian has the block-arrow structure

$$\mathbf{F}_t = \begin{bmatrix} \mathbf{A}_t & \mathbf{B}_t^{(1)} & \mathbf{B}_t^{(2)} & \cdots & \mathbf{B}_t^{(L)} \\ \mathbf{C}_t^{(1)} & \mathbf{D}_t^{(1)} & 0 & \cdots & 0 \\ \mathbf{C}_t^{(2)} & 0 & \mathbf{D}_t^{(2)} & \cdots & 0 \\ \vdots & \vdots & \vdots & \ddots & \vdots \\ \mathbf{C}_t^{(L)} & 0 & 0 & \cdots & \mathbf{D}_t^{(L)} \end{bmatrix}.$$

Here $\mathbf{A}_t$ encodes voltage–voltage coupling, $\mathbf{B}_t^{(\ell)}$ and $\mathbf{C}_t^{(\ell)}$ capture voltage–gate interactions, and $\mathbf{D}_t^{(\ell)}$ govern gate self-dynamics. In biophysical HH models, $\mathbf{C}_t^{(\ell)}$ and $\mathbf{D}_t^{(\ell)}$ are diagonal, and $\mathbf{B}_t^{(\ell)}$ are nearly diagonal (apart from weak solver-induced mixing). We assume that these off-diagonal elements are negligible and consider in the following that $\mathbf{B}_t^{(\ell)}$ are diagonal. The main source of non-diagonality lies in $\mathbf{A}_t$, through axial coupling between compartments.

***Magnitude of cross-derivatives.*** We examine the relative size of the partial derivatives that make up $\mathbf{F}_t$. In the Hodgkin–Huxley model, these correspond to four types of dependencies: (i) gating variable on voltage ($\partial x_{t+1}/\partial V_t$), (ii) gating variable on gating variable ($\partial x_{t+1}/\partial x_t$), (iii) voltage on gating variable ($\partial V_{t+1}/\partial x_t$), and (iv) voltage on voltage. We analyze each in turn.

*(i) Gate with respect to voltage.* Consider a generic gating variable $x$ governed by

$$\dot{x} = \alpha(V)(1-x) - \beta(V)x,$$

where $\alpha(V)$ and $\beta(V)$ are voltage-dependent transition rates. The exponential–Euler update gives:

$$x_{t+1} = x_\infty(V_t) + \left( x_t - x_\infty(V_t) \right) e^{-\Delta t/\tau(V_t)}, \tag{11}$$

with

$$x_\infty(V) = \frac{\alpha(V)}{\alpha(V) + \beta(V)}, \quad \tau(V) = \frac{1}{\alpha(V) + \beta(V)}.$$

Differentiating $x_{t+1}$ with respect to $V_t$ gives:

$$\frac{\partial x_{t+1}}{\partial V_t} = x'_\infty(V_t)\left( 1 - e^{-\Delta t/\tau(V_t)} \right) + (x_t - x_\infty(V_t))\, k'(V_t),$$

where $k(V_t) = e^{-\Delta t/\tau(V_t)}$. It follows that (using the chain rule on $k$):

$$x'_\infty(V_t) = \frac{\alpha'(V_t)\beta(V_t) - \alpha(V_t)\beta'(V_t)}{[\alpha(V_t) + \beta(V_t)]^2}, \quad k'(V_t) = -\Delta t\, (\alpha'(V_t) + \beta'(V_t))\, e^{-\Delta t(\alpha(V_t)+\beta(V_t))}.$$

Applying the triangle inequality and using that $|x_t - x_\infty| \leq 1$ (since $x \in [0,1]$) yields the bound

$$\left| \frac{\partial x_{t+1}}{\partial V_t} \right| \leq (|\alpha'| + |\beta'|) \left[ \frac{1 - e^{-\Delta t(\alpha+\beta)}}{\alpha + \beta} + \Delta t\, e^{-\Delta t(\alpha+\beta)} \right].$$

For small $\Delta t$, with moderate and smooth transition rates $\alpha, \beta$, this is of order $\mathcal{O}(\Delta t)$, confirming that voltage-to-gate sensitivities are small under typical HH kinetics.

*(ii) Gate self-derivative.* It follows directly from eq. (11) that:

$$\frac{\partial x_{t+1}}{\partial x_t} = e^{-\Delta t \, (\alpha(V_t) + \beta(V_t))} \in (0, 1],$$

so it is $\mathcal{O}(1)$ and typically much larger than the cross term $\partial x_{t+1}/\partial V_t$ when $\Delta t$ is small and $|\alpha'|, |\beta'|$ are moderate.

*(iii) Voltage with respect to gate.* Next, consider the effect of a gating variable $x_t$ on the voltage update. In practice, JAXLEY uses an implicit solver. Here for the sake of clarity, we assume an explicit Euler step. For a fixed compartment, it can be written as:

$$V_{t+1} = V_t + \frac{\Delta t}{C_m} \left[ \frac{I_{\text{ext}}}{S} - \sum_c g_c \, a_c(t) \, (V_t - E_c) + G_{\text{ax}}(\mathbf{V}_t) \right], \tag{12}$$

where $a_c(t)$ is the activation of channel $c$. For the channel $c^\star$ corresponding to gate $x_t$, the only dependence is through $a_{c^\star}(x_t)$, giving

$$\frac{\partial V_{t+1}}{\partial x_t} = -\frac{\Delta t}{C_m} g_{c^\star} \, a'_{c^\star}(x_t) \, (V_t - E_{c^\star}).$$

In practice, $a_{c^\star}(x_t) = C x_t^k$ with $k \in \{1, \ldots, 4\}$ and $C, x_t \in [0, 1]$. Indeed recall that the activation is a product of gating variables (between 0 and 1) to some integer power. Then $|a'_{c^\star}(x_t)| \leq kC \leq k$. Thus,

$$\left| \frac{\partial V_{t+1}}{\partial x_t} \right| \lesssim k \, \Delta t \, g_{c^\star} \, |V_t - E_{c^\star}|.$$

With $\Delta t = 0.025$ ms, $g_{c^\star} \lesssim 1 \, \text{S/cm}^2$, and $|V_t - E_{c^\star}| \lesssim 100$ mV, these terms are the dominant off-diagonal contribution to $\mathbf{F}_t$ becoming large primarily during spikes. Provided that spikes do not occur at high frequency throughout the cell, these contributions remain intermittent and should not overwhelm the diagonal structure of $\mathbf{F}_t$.

*(iv) Voltage with respect to voltage.* In eq. (12), the voltage of compartment $i$ is coupled to the voltage of its neighboring compartments via the axial conductance term:

$$G_{\text{ax}}^{(i)}(\mathbf{V}_t) = \sum_{j \sim i} g^{(i,j)} \, (V_t^{(j)} - V_t^{(i)}).$$

It follows that

$$\frac{\partial V_{t+1}^{(i)}}{\partial V_t^{(j)}} = \begin{cases} 1 + \frac{\Delta t}{C_m} \left[ -\sum_c g_c \, a_c(t) - \sum_k g^{(i,k)} \right], & j = i, \\ \frac{\Delta t}{C_m} g^{(i,j)}, & j \neq i. \end{cases}$$

Thus, diagonal entries are $\mathcal{O}(1)$ (since the activations $|a_c(t)| \leq 1$), while off-diagonal ones scale as $\mathcal{O}(\Delta t)$.

### What conditions support the diagonal approximation?

The preceding derivative analysis shows that, for a broad class of *well-behaved* Hodgkin-Huxley models and sufficiently small time steps $\Delta t$, the state transition Jacobian $\mathbf{F}_t$ exhibits limited cross-coupling between state variables. In particular, most off-diagonal Jacobian entries scale as $\mathcal{O}(\Delta t)$, while the diagonal entries remain $\mathcal{O}(1)$. The main exception is given by the voltage-gate sensitivities $\partial V_{t+1}/\partial x_t$, which may not be negligible during action potentials.

We refer to a model as *well-behaved* when the following conditions hold:

- The gating rate functions $\alpha(V), \beta(V)$ and their derivatives remain bounded over the relevant voltage range, ensuring that gate-voltage and voltage-voltage cross-derivatives scale as $\mathcal{O}(\Delta t)$.

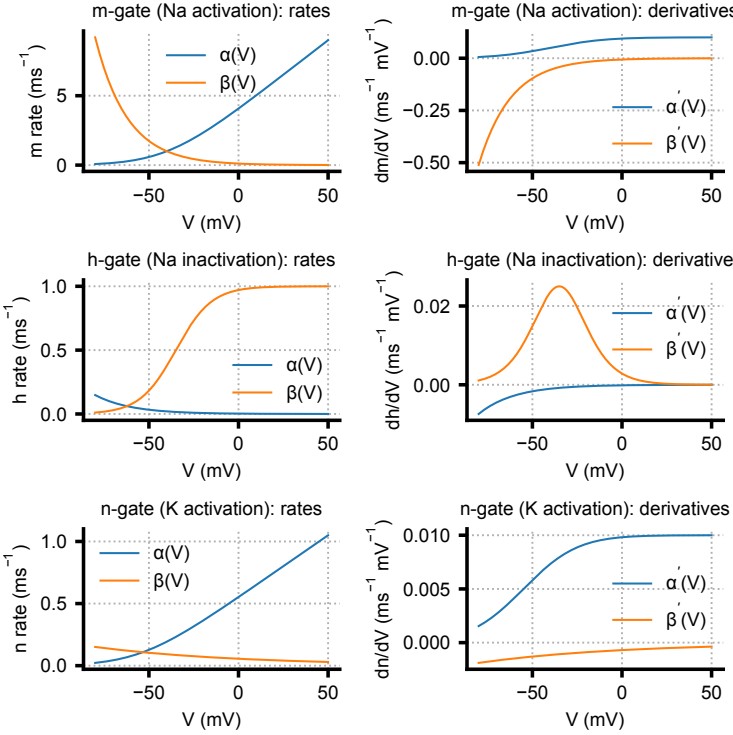

Figure B.1: **Rate functions (left) and their derivatives (right) over plausible values of the membrane voltage for the standard HH model [6].** The rate functions and their derivatives are bounded.

- Spiking activity is not excessively frequent, so that large voltage-gate sensitivities do not persist over time.

Under these conditions, all sources of non-diagonality in $\mathbf{F}_t$ are negligible for small $\Delta t$, except for the voltage-gate derivatives. Standard Hodgkin-Huxley parameterizations typically satisfy the first condition: the transition rates and their derivatives are smooth and moderate in magnitude, and choosing $\Delta t$ sufficiently small ensures that most off-diagonal Jacobian entries remain negligible relative to the diagonal terms (see fig. B.1).

The dominant deviation from diagonality arises from the voltage-gate terms $\partial V_{t+1}/\partial x_t$. During spikes, these derivatives can become large; at lower firing rates they are typically smaller, but can still remain of the same order of magnitude as the main diagonal terms (fig. B.2). Consequently, while well-behaved dynamics eliminate most sources of cross-coupling, the diagonal approximation cannot be justified from Jacobian structure alone in the presence of spiking.

Nevertheless, even when voltage-gate sensitivities are large, their impact on the predicted covariance can be controlled through the variances assigned to the gating states. In the EKF prediction step, the covariance update is given by

$$\mathbf{\Sigma}_{t|t-1} = \mathbf{F}_t\mathbf{\Sigma}_{t-1|t-1}\mathbf{F}_t^\top + \mathbf{Q}_t,$$

with diagonal prior covariance $\mathbf{\Sigma}_{t-1|t-1} = \mathrm{diag}(s_1, \ldots, s_d)$. The off-diagonal entries of $\mathbf{\Sigma}_{t|t-1}$ arise from terms of the form $\mathbf{F}_{t,ik}s_k\mathbf{F}_{t,jk}$ for $i \neq j$. Assigning small process and initial variances to gating variables directly suppresses these contributions, even when the corresponding Jacobian entries are non-negligible. This choice is also physically well motivated: gating variables represent channel states confined to the interval $[0, 1]$, so their intrinsic uncertainty is naturally much smaller than that of membrane voltages. This procedure allows to maintain the validity of the diagonal covariance approximation.

The validity of the diagonal EKF now heavily relies on the emission model, and more specifically on whether a diagonal approximation can be made for $\mathbf{H}_t^\top\mathbf{R}_t^{-1}\mathbf{H}_t$.

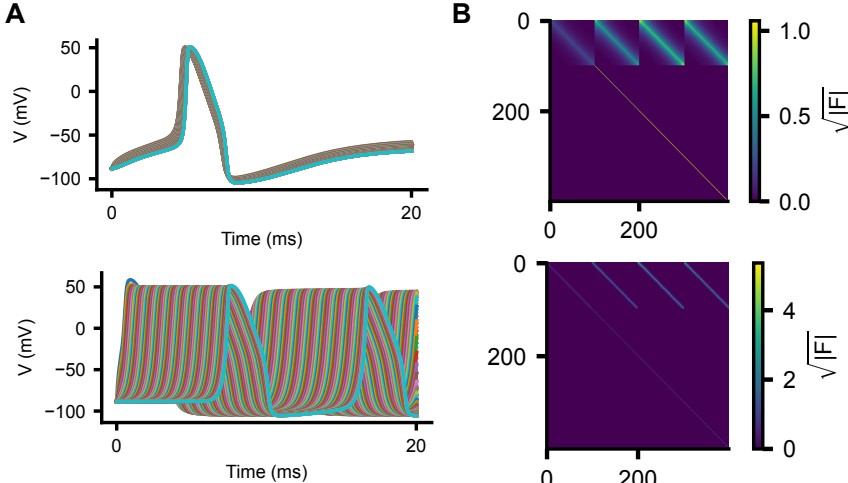

Figure B.2: **Structure of the dynamics Jacobian for a 100-compartment HH cable ($\Delta t = 0.025$ ms). A:** Simulated membrane voltages for a low-frequency spiking regime (top) and a high-frequency regime (bottom). **B:** Square-root–scaled heatmaps of the dynamics Jacobian averaged over time in both regimes. Consistent with our analysis, gate-to-voltage ($\partial x_{t+1}/\partial V_t$) and voltage–neighbor ($\partial V_{t+1}^{(i)}/\partial V_t^{(j)}$) couplings remain weak relative to the diagonal terms. The main distinction between regimes is that voltage-to-gate sensitivities ($\partial V_{t+1}/\partial x_t$) increase substantially during high-frequency spiking, overwhelming the diagonal entries. At lower firing rates, these terms are often smaller, yet can remain comparable to the main diagonal terms.

—

### Diagonality of $\mathbf{H}_t^\top \mathbf{R}_t^{-1} \mathbf{H}_t$

In our work, we assumed whitened observation noise, $\mathbf{R}_t = \sigma_{\text{obs}}^2 \mathbf{I}$, so diagonality depends solely on $\mathbf{H}_t^\top \mathbf{H}_t$. Let the first $K$ state dimensions correspond to compartment voltages and the remainder to gating variables. Since observations depend only on voltages,

$$\mathbf{H}_t = \begin{bmatrix} \mathbf{H}_t^v & 0 \end{bmatrix}, \quad \mathbf{H}_t^v \in \mathbb{R}^{M \times K},$$

so that

$$\mathbf{H}_t^\top \mathbf{H}_t = \begin{bmatrix} (\mathbf{H}_t^v)^\top \mathbf{H}_t^v & 0 \\ 0 & 0 \end{bmatrix}.$$

Crucially, extracellular potentials are linear in membrane voltages (cf. eq. (4)). Because this mapping is static, $\mathbf{H}_t^v = \mathbf{H}^v$ for all $t$. The diagonality of $(\mathbf{H}^v)^\top \mathbf{H}^v$ therefore depends only on the spatial arrangement of electrodes and compartments.

Let's analyze the structure of $\mathbf{H}^v$. Recall that the transmembrane current in compartment $i$ is

$$I_m^{(i)} = I_{\text{ext}}^{(i)} + S^{(i)} \sum_{j \sim i} g^{(i,j)}(V^{(j)} - V^{(i)}),$$

and electrode $m$ measures

$$\Phi_m = \sum_{i=1}^N \frac{\kappa}{r_{mi}} I_m^{(i)},$$

where $r_{mi}$ is the electrode–compartment distance and $\kappa$ depends on properties of the extracellular medium. Differentiating yields the "self + neighbors" structure:

$$H_{mj}^v = -\frac{\kappa}{r_{mj}} S^{(j)} \sum_{k \sim j} g^{(j,k)} + \sum_{i: i \sim j} \frac{\kappa}{r_{mi}} S^{(i)} g^{(i,j)}.$$

Define the inverse–distance matrix $R \in \mathbb{R}^{M \times N}$ with $R_{mi} = \kappa / r_{mi}$, a diagonal matrix $D$ with

$$D_{jj} = S^{(j)} \sum_{k \sim j} g^{(j,k)},$$

and an axial coupling matrix $G$ with

$$G_{ij} = \begin{cases} S^{(i)} g^{(i,j)}, & i \sim j, \\ 0, & \text{otherwise.} \end{cases}$$

Then

$$\mathbf{H}^v = R(G - D), \qquad (\mathbf{H}^v)^\top \mathbf{H}^v = (G - D)^\top W (G - D),$$

where $W := R^\top R$.

Let $C := G - D$ and denote its entries by $c_{pi}$; the support of column $i$ is the local neighborhood $S_i = \{i\} \cup \{k : k \sim i\}$. Then

$$\left[ (\mathbf{H}^v)^\top \mathbf{H}^v \right]_{ij} = \sum_{p \in S_i} \sum_{q \in S_j} c_{pi} \, c_{qj} \, W_{pq}.$$

The electrode Gram matrix is

$$W_{pq} = \sum_{m=1}^{M} \frac{\kappa^2}{r_{mp} r_{mq}}. \tag{13}$$

Using (13), $W_{pq}$ becomes large when electrodes are simultaneously close to compartments $p$ and $q$. Therefore,

$$\left[ (\mathbf{H}^v)^\top \mathbf{H}^v \right]_{ij}$$

grows when both neighborhoods $S_i$ and $S_j$ lie within the sensitive region of multiple electrodes. This regime is typical for high-density probes such as Neuropixels or MEAs, where many electrodes lie close to many compartments, yielding strong off–diagonal structure in $(\mathbf{H}^v)^\top \mathbf{H}^v$ and limiting the accuracy of the diagonal EKF (cf. Section 5.1).

**When is $\mathbf{H}^{v\top} \mathbf{H}^v$ diagonal?** When no electrode is simultaneously close to two different compartments, $W$ is approximately diagonal and

$$c_i^\top W c_j \approx \sum_{p \in S_i \cap S_j} c_{pi} \, c_{pj} \, W_{pp}.$$

If the overlap $S_i \cap S_j$ is small, the off–diagonal terms remain strongly attenuated. In this regime, $(\mathbf{H}^v)^\top \mathbf{H}^v$ is close to diagonal, and a diagonal EKF update accurately captures the dynamics covariance.

A concrete example is a linear branch with $N$ compartments of length $L$, and $N$ electrodes placed so that electrode $i$ lies at distance $l \ll L$ from compartment $i$. Each electrode then primarily measures one compartment, yielding nearly diagonal $W$. Since a linear cable has minimal neighborhood overlap ($S_i \cap S_j = \emptyset$ unless $j = i \pm 1$), $(\mathbf{H}^v)^\top \mathbf{H}^v$ becomes tridiagonal with dominant diagonal entries. This setting aligns with our single-branch experiment and explains why the diagonal EKF performs well there (Section 4.1).

A similar argument applies in the multi-branch example (Section 4.2). The compartments in each branch are sufficiently long that electrodes are rarely simultaneously close to two distinct compartments. As a result, $W$ remains approximately diagonal, and $(\mathbf{H}^v)^\top \mathbf{H}^v$ preserves a near-tridiagonal structure, with the only substantial off-diagonal contributions occurring near the branch point where neighborhood overlap increases. Although the diagonal approximation is less accurate than in the single-branch case, our results show that it still achieves good reconstruction performance in this geometry.

***Takeaway.*** For *well-behaved* Hodgkin-Huxley models, the state transition Jacobian $\mathbf{F}_t$ exhibits limited cross-coupling, with most off-diagonal entries suppressed by $\Delta t$. The primary exception arises from voltage-gate sensitivities, which can become comparable to diagonal terms during spiking. Nevertheless, because off-diagonal contributions to the predicted covariance are weighted by the variances of the corresponding state variables, assigning small variances to gating states—naturally

justified by their bounded dynamics—allows the predicted covariance to remain approximately diagonal in practice. In contrast, the problem geometry may cause $\mathbf{H}_t^\top \mathbf{R}_t^{-1} \mathbf{H}_t$ to deviate strongly from diagonality, which can undermine the validity of a diagonal filtered covariance approximation.

***Limitation.*** The above argument for the predicted step is heuristic and intended to motivate when a diagonal approximation of the predicted covariance is reasonable. In practice, solver-induced coupling between compartment voltages can introduce additional non-diagonality, particularly for morphologies of increasing complexity with many interconnected compartments. While this effect may reduce the accuracy of the diagonal approximation in such settings, it was not observed to be limiting in the experimental regimes considered in this work.

## C   Mathematical details of the Block-Diagonal EKF

### C.1   Definition and closedness

We now describe the EKF when covariances are constrained to a block-diagonal form. We assume that the first $K$ latent dimensions correspond to compartment voltages and the remaining ones to gating variables. At each time step $t$, we assume that both predicted and filtered covariances take the form

$$\mathbf{\Sigma}_{t\,|\,\star} = \begin{bmatrix} \mathbf{\Sigma}_{t\,|\,\star}^{vv} & 0 \\ 0 & \mathrm{diag}(\boldsymbol{\sigma}_{t\,|\,\star}^2) \end{bmatrix}, \qquad \star \in \{t-1, t\},$$

where $\mathbf{\Sigma}_{t\,|\,\star}^{vv} \in \mathbb{R}^{K \times K}$ is dense and $\boldsymbol{\sigma}_{t\,|\,\star}^2$ contains the gating variances. We denote this family of matrices by $\mathrm{BD}(K)$.

**Proposition 1** (Closedness of the BD($K$) family). *Assume that for all $t$:*

1. *The dynamics covariance $\mathbf{Q}_t$ is BD(K).*

2. *The dynamics Jacobian $\mathbf{F}_t$ is BD(K).*

*If $\mathbf{\Sigma}_0$ is BD($K$), then both predicted and filtered covariances remain BD($K$) for all subsequent time steps.*

*Proof of Proposition 1.* Writing $\mathbf{z}_t = [\mathbf{v}_t; \mathbf{g}_t]$, assume $\mathbf{\Sigma}_{t-1|t-1}$ is BD($K$):

$$\mathbf{\Sigma}_{t-1|t-1} = \begin{bmatrix} \mathbf{\Sigma}_{t-1|t-1}^{vv} & 0 \\ 0 & \mathrm{diag}(\boldsymbol{\sigma}_{t-1|t-1}^2) \end{bmatrix}.$$

Under Assumptions 1–2, $\mathbf{F}_t = \mathrm{blkdiag}(\mathbf{A}_t, \mathrm{diag}(\mathbf{d}_t))$ and $\mathbf{Q}_t$ share this structure. The EKF prediction $\mathbf{\Sigma}_{t|t-1} = \mathbf{F}_t \mathbf{\Sigma}_{t-1|t-1} \mathbf{F}_t^\top + \mathbf{Q}_t$ then yields

$$\mathbf{\Sigma}_{t|t-1} = \begin{bmatrix} \mathbf{A}_t \mathbf{\Sigma}_{t-1|t-1}^{vv} \mathbf{A}_t^\top + \mathbf{Q}_t^{vv} & 0 \\ 0 & \mathrm{diag}(\mathbf{d}_t^2 \odot \boldsymbol{\sigma}_{t-1|t-1}^2 + \mathbf{q}_t^2) \end{bmatrix}.$$

For the update step, we use the identity:

$$\mathbf{\Sigma}_{t|t} = \left( \mathbf{\Sigma}_{t|t-1}^{-1} + \mathbf{H}_t^\top \mathbf{R}_t^{-1} \mathbf{H}_t \right)^{-1}.$$

Now,

$$\mathbf{\Sigma}_{t|t-1} = \begin{bmatrix} \mathbf{\Sigma}_{t|t-1}^{vv} & 0 \\ 0 & \mathrm{diag}(\boldsymbol{\sigma}_{t|t-1}^2) \end{bmatrix} \quad \Rightarrow \quad \mathbf{\Sigma}_{t|t-1}^{-1} = \begin{bmatrix} (\mathbf{\Sigma}_{t|t-1}^{vv})^{-1} & 0 \\ 0 & \mathrm{diag}\left((\boldsymbol{\sigma}_{t|t-1}^2)^{-1}\right) \end{bmatrix}.$$

Since $\mathbf{H}_t = [\mathbf{H}_t^v\ 0]$,

$$\mathbf{H}_t^\top \mathbf{R}_t^{-1} \mathbf{H}_t = \begin{bmatrix} \mathbf{J}_t^v & 0 \\ 0 & 0 \end{bmatrix}, \qquad \mathbf{J}_t^v := \mathbf{H}_t^{v\top} \mathbf{R}_t^{-1} \mathbf{H}_t^v.$$

Therefore the sum inside the inverse is block diagonal,

$$\mathbf{\Sigma}_{t|t-1}^{-1} + \mathbf{H}_t^\top \mathbf{R}_t^{-1} \mathbf{H}_t = \begin{bmatrix} (\mathbf{\Sigma}_{t|t-1}^{vv})^{-1} + \mathbf{J}_t^v & 0 \\ 0 & \mathrm{diag}\left((\boldsymbol{\sigma}_{t|t-1}^2)^{-1}\right) \end{bmatrix},$$

and inverting yields

$$\boldsymbol{\Sigma}_{t|t} = \begin{bmatrix} \left((\boldsymbol{\Sigma}_{t|t-1}^{vv})^{-1} + \mathbf{J}_t^v\right)^{-1} & 0 \\ 0 & \mathrm{diag}(\boldsymbol{\sigma}_{t|t-1}^2) \end{bmatrix}.$$

Hence both predicted and filtered covariances remain BD($K$). Note that only the voltage block is updated.

Let's derive the update for the filtered mean. We use:

$$\boldsymbol{\mu}_{t|t} = \boldsymbol{\mu}_{t|t-1} + \boldsymbol{\Sigma}_{t|t}\,\mathbf{H}_t^\top \mathbf{R}_t^{-1}\big(\mathbf{y}_t - \mathbf{h}_t(\boldsymbol{\mu}_{t|t-1})\big).$$

Let the residual be

$$\mathbf{r}_t := \mathbf{y}_t - \mathbf{h}_t(\boldsymbol{\mu}_{t|t-1}).$$

Since $\mathbf{H}_t = [\mathbf{H}_t^v\ 0]$ and

$$\boldsymbol{\Sigma}_{t|t} = \begin{bmatrix} \boldsymbol{\Sigma}_{t|t}^{vv} & 0 \\ 0 & \mathrm{diag}(\boldsymbol{\sigma}_{t|t-1}^2) \end{bmatrix}, \quad \mathbf{H}_t^\top \mathbf{R}_t^{-1}\mathbf{r}_t = \begin{bmatrix} \mathbf{H}_t^{v\top}\mathbf{R}_t^{-1}\mathbf{r}_t \\ 0 \end{bmatrix},$$

we obtain the block update

$$\boldsymbol{\mu}_{t|t} = \begin{bmatrix} \boldsymbol{\mu}_{t|t-1}^v \\ \boldsymbol{\mu}_{t|t-1}^g \end{bmatrix} + \begin{bmatrix} \boldsymbol{\Sigma}_{t|t}^{vv}\,\mathbf{H}_t^{v\top}\mathbf{R}_t^{-1}\mathbf{r}_t \\ 0 \end{bmatrix} = \begin{bmatrix} \boldsymbol{\mu}_{t|t-1}^v + \boldsymbol{\Sigma}_{t|t}^{vv}\,\mathbf{H}_t^{v\top}\mathbf{R}_t^{-1}\mathbf{r}_t \\ \boldsymbol{\mu}_{t|t-1}^g \end{bmatrix}.$$

Thus only the first $K$ states (the voltages) are updated; the gating variables remain unchanged, which is consistent with the observation depending solely on the voltage block.

$$\square$$

This structure reduces the EKF complexity from $\mathcal{O}(d^3)$ time and $\mathcal{O}(d^2)$ memory to $\mathcal{O}(K^3 + d)$ and $\mathcal{O}(K^2 + d)$, respectively, a major gain since $K \ll d$ in typical large neuron models.

—

## C.2   Validity of the block-diagonal approximation

We now discuss the assumptions of Proposition 1.

*Assumption 1 (BD($K$) process noise).* In our setting, process noise is injected independently per compartment and gating variable, so $\mathbf{Q}_t$ is naturally BD($K$) since it is diagonal.

*Assumption 2 (BD($K$) Jacobian).* The exact Jacobian is not strictly BD($K$) but has the arrow-shaped form

$$\mathbf{F}_t = \begin{bmatrix} \mathbf{A}_t & \mathbf{B}_t^{(1)} & \mathbf{B}_t^{(2)} & \cdots & \mathbf{B}_t^{(L)} \\ \mathbf{C}_t^{(1)} & \mathbf{D}_t^{(1)} & 0 & \cdots & 0 \\ \mathbf{C}_t^{(2)} & 0 & \mathbf{D}_t^{(2)} & \cdots & 0 \\ \vdots & \vdots & \vdots & \ddots & \vdots \\ \mathbf{C}_t^{(L)} & 0 & 0 & \cdots & \mathbf{D}_t^{(L)} \end{bmatrix}, \tag{14}$$

where $\mathbf{A}_t$ captures voltage–voltage coupling and the other blocks represent voltage–gate interactions and self-dynamics. Under the BD($K$) approximation, we set

$$\mathbf{B}_t^{(\ell)} = \mathbf{C}_t^{(\ell)} = 0, \quad \forall \ell,$$

yielding

$$\mathbf{F}_t = \mathrm{blkdiag}\big(\mathbf{A}_t, \mathbf{D}_t^{(1)}, \ldots, \mathbf{D}_t^{(L)}\big), \qquad \mathbf{D}_t^{(\ell)} \text{ diagonal.}$$

This corresponds to neglecting cross-sensitivities between voltages and gates, i.e. assuming $\partial V_{t+1}/\partial x_t \approx 0$ and $\partial x_{t+1}/\partial V_t \approx 0$ for each gate $x$.

As established in Appendix B, the gate-to-voltage terms ($\partial x_{t+1}/\partial V_t$) are negligible under well-behaved HH dynamics. Voltage-to-gate sensitivities ($\partial V_{t+1}/\partial x_t$) grow primarily during frequent

spiking activity and can become significant in those regimes, although they often remain manageable when spiking rates are low. When these terms become problematic, the same mitigation strategy used for the diagonal approximation applies: assigning very small process and initial variances to gating states suppresses their contribution to the predicted covariance.

Under these conditions, a block-diagonal covariance structure remains reasonable while still supporting dense inter-compartmental voltage coupling through $\mathbf{A}_t$.

## D   Unidentifiability of 3D compartment locations

In this section, we show that when inferring a neuron's spatial position from extracellular recordings, the geometry of the recording sites can impose inherent unidentifiability in the recovered compartment locations.

Our multi-compartment model represents the neuron as a collection of compartments, each treated as a point source of transmembrane current located at its center. Hence, estimating a neuron's spatial configuration reduces to estimating the 3D positions of its compartments. According to eq. (4), the extracellular potential measured at a given electrode depends only on the transmembrane currents and the inverse distance to each compartment. Therefore, if the distances between each compartment and all electrodes remain unchanged, the resulting extracellular signals will be identical.

We first consider the case where all electrodes lie on a single line (fig. D.1A). Fix a single compartment at some position in 3D space (black). The set of all locations that preserve its distance to a given site is a sphere. The intersection of these spheres—corresponding to multiple sites aligned on a line—is a circle. Thus, any position on this circle (blue dashed circle) yields the same set of distances (and hence the same extracellular potentials), rendering the compartment's position unidentifiable up to a rotation around the axis defined by the recording sites.

Next, suppose the electrodes lie on two parallel lines, forming a planar grid (fig. D.1B). In this case, for a given compartment (black), the set of equidistant locations from the first line of sites lies on a circle (blue), and similarly for the second line (green). The only positions that simultaneously preserve distances to both sets of sites lie at the intersection of these two circles. This intersection consists of exactly two points: the true location of the compartment and its mirror reflection across the plane defined by the electrodes (grey).

This ambiguity persists even with more than two lines of electrodes, as long as they all lie in the same plane. For example, fig. D.1C shows three parallel lines of recording sites in a planar grid, viewed from above. The mirror-reflected location remains the only position (besides the true one) that produces identical distances to all recording sites.

In general, if all electrodes lie in a 2D plane (e.g., $z = 0$), then for any compartment, the reflection of its true location across this plane yields the same distances to all electrodes. Consequently, the extracellular voltages at all sites are unchanged. This holds independently of the number or spacing of electrodes and applies to all compartments of a multi-compartment model. Therefore, in the absence of additional constraints, the entire neuron morphology is identifiable only up to reflections of its compartments across the recording plane.

This ambiguity can be partially mitigated by incorporating structural constraints or prior knowledge about the neuron. For example, if the cell is known to consist of a single linear branch, then the configuration of all compartments is restricted to lie along a line. In this case, the full morphology is only ambiguous up to a reflection of that line across the recording plane—substantially reducing the space of indistinguishable configurations. Similarly, if the compartmental connectivity is known (i.e., which compartments are connected to each other and in what topology), then only configurations that preserve both the pairwise distances to electrodes and the known connectivity are allowed. While this does not fully resolve the mirror symmetry, it rules out arbitrary spatial rearrangements. Finally, full identifiability can be achieved by breaking the planar symmetry of the recording sites. For example, recordings from two probes placed at different depths or orientations—such that their electrodes do not lie in the same plane—remove the reflection ambiguity and enable unique localization of each compartment in 3D space.

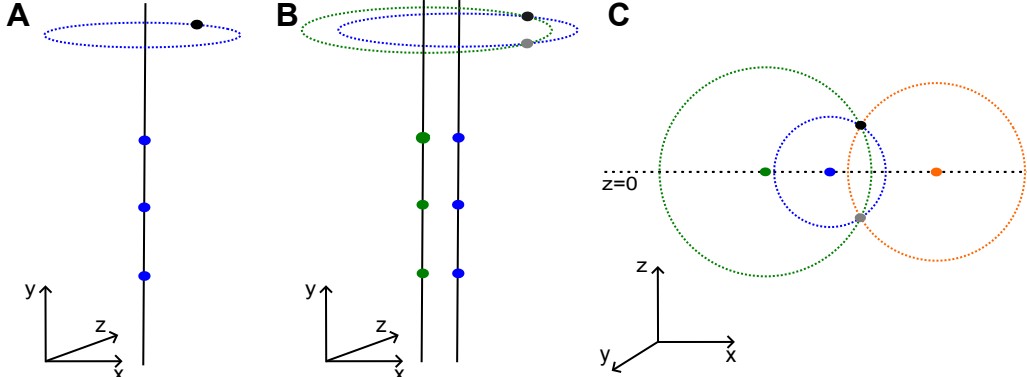

Figure D.1: **Unidentifiability of the positions. A**: The distance between the compartment (black) and each electrode (blue) is unchanged as it moves along the blue dashed circle. It follows that any position of the compartment on the circle yields the same observations at all the electrodes. **B**: Compartment's distance to each blue (resp. green) electrode is unchanged as it moves along the blue (resp. green) dashed circle. Hence, the grey location is the only other position that yields the same observations as the black one at all electrodes. **C**: Panel **B** viewed from above with now *three* parallel lines containing electrodes (green, blue, orange ones). Since all the electrodes lie in a 2D-plane ($z = 0$), the only position that yields the same observations as the true location at each electrode is the true location's reflection with respect to the plane $z = 0$.

# E    Additional Details and Figures for Section 4

This section provides additional experimental details, biophysical parameters, and figures related to the validation experiments presented in Section 4.

The extracellular resistivity was assumed to be homogeneous and equal to $300\,\Omega \cdot \mathrm{cm}$, and the specific membrane capacitance was fixed to $C_m = 1\,\mu\mathrm{F}/\mathrm{cm}^2$ across all compartments. While we assume homogeneous extracellular resistivity, this simplification neglects known spatial variability [25], which could be addressed in future work. Extracellular voltages were recorded at a sampling rate of $40\,\mathrm{kHz}$.

All voltages and channel states were initialized at their steady-state values at rest, and thus we assumed the initial state to be known in the EKF. We set the initial covariance matrix of the EKF to $\boldsymbol{\Sigma}_0 = 0.1\mathbf{I}$. Observation noise was also assumed to be known, as commercial devices that record extracellular potentials typically provide noise specifications [16, 38].

In Sections 4.1 and 4.2, where the state dynamics were well-specified and deterministic, we fixed the standard deviation of the membrane potentials and channel states to small values: $\sigma_v = 0.0001\,\mathrm{mV}/\mathrm{ms}^{1/2}$, and $\sigma_{\mathrm{state}} = 0.00001\,\mathrm{ms}^{-1/2}$ (same for all the channels) for all compartments. These values ensured numerical stability while avoiding degenerate gradients during optimization.

## E.1    Validation on single-branch model

We evaluated a single-branch cell model as a baseline for benchmarking. This branch consists of 10 cylindrical compartments of length $24\,\mu\mathrm{m}$ and radius $2\,\mu\mathrm{m}$, equipped with standard Hodgkin–Huxley (HH) sodium, potassium, and leak channels [6]. The maximum conductances and reversal potentials are provided in Table 1. A constant current of amplitude $1.5\,\mathrm{nA}$ was injected for $20\,\mathrm{ms}$.

As the neuron–electrode distance increased, the signal-to-noise ratio (SNR) decreased, resulting in higher bias and variance in inferred conductances (fig. E.1B). At short distances with high SNR, we observed that MSE minimization, standard EKF and block-diagonal EKF yielded comparable estimates, indicating that EKF's linearization introduced minimal error and confirming the validity of the block-diagonal approximation (fig. E.1C). The diagonal EKF introduces a slightly larger bias, but its voltage reconstructions remain highly accurate (fig. E.2). This is consistent with our analysis

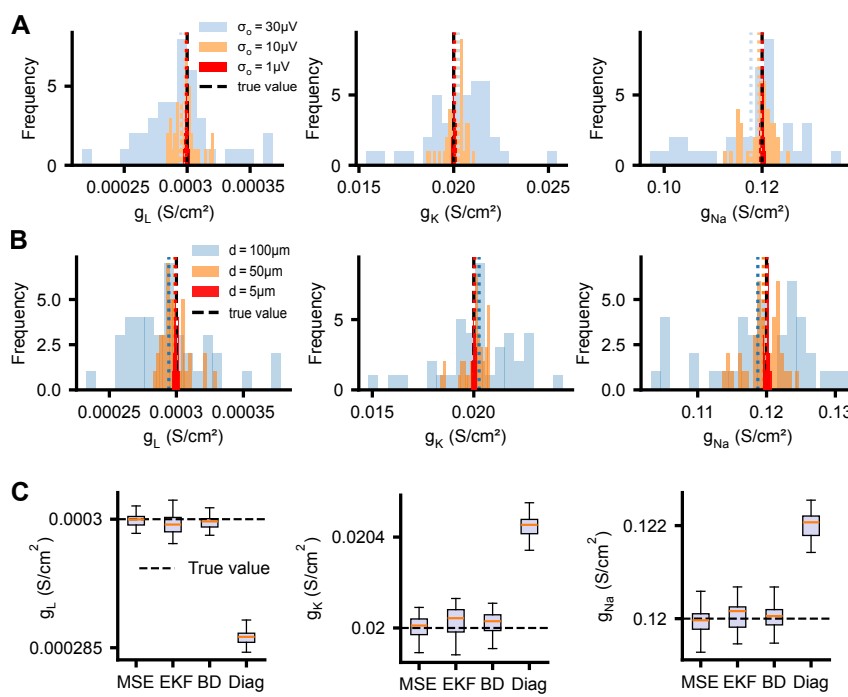

Figure E.1: **Validation on synthetic single-branch. A**: Distributions of inferred maximum conductances under different observation noise levels $\sigma_{obs}$ (colored dashed lines indicate the means). **B**: Distributions of inferred maximum conductances for different cell-probe distances (colored dashed lines indicate the means). **C**: Distributions of inferred maximum conductances for various inference methods with $\sigma_{obs} = 1\,\mu V$ and $d = 5\mu m$ (orange lines indicate the means).

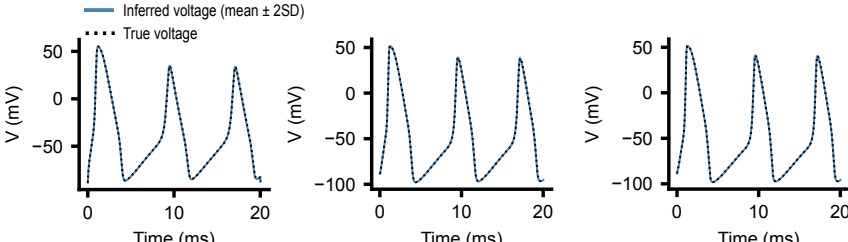

Figure E.2: **Recovery of membrane voltage with diagonal EKF in the single-branch model for compartments 1 (left), 5 (center) and 10 (right).** The mean and standard deviation are estimated by considering 40 randomly simulated datasets at fixed observation noise $\sigma_{obs} = 1\mu V$. Despite a small bias in the estimation of the parameters, the voltage traces are still reconstructed accurately.

|  | Na | K | Leak |
|---|---|---|---|
| Maximum conductance (S/cm$^2$) | 0.12 | 0.02 | 0.003 |
| Reversal potential (mV) | 53.0 | $-107.0$ | $-88.5188$ |

Table 1: Biophysical parameters of the single-branch model.

in Appendix B: only the first (soma) compartment lies near the electrode, with all others much farther away, so cross-visibility is minimal and a diagonal covariance approximation is sufficient. As observation noise increased, estimation accuracy declined (fig. E.1A), but even at $30\,\mu V$ RMS—a challenging noise regime (fig. E.3C)—estimates remained close to the ground truth. Notably, the

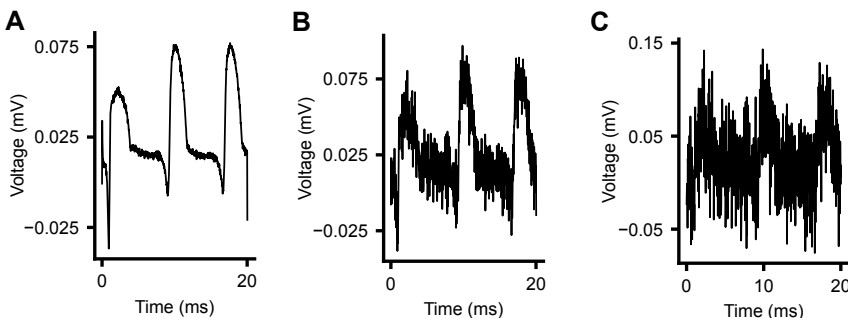

Figure E.3: **Extracellular voltage traces simulated for different noise levels in the single-branch model. A**: $\sigma_{obs} = 1\mu V$, **B**: $\sigma_{obs} = 10\mu V$, **C**: $\sigma_{obs} = 30\mu V$ represents a challenging noise regime. The electrode is $5\mu m$ away from the cell.

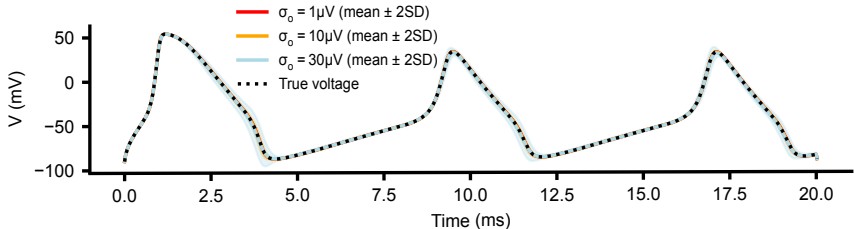

Figure E.4: **Recovery of membrane voltage for different observation noise levels in the single-branch model.** The mean and standard deviation are estimated by considering 40 datasets simulated at fixed observation noise. Even in challenging noise regimes ($\sigma_{obs} = 30\mu V$), the induced variance on the voltage trace reconstruction is negligible.

induced variance on the reconstructed voltage traces is small (fig. E.4), indicating robustness even under challenging conditions.

### E.2 Validation on multi-branch neuron model

We also evaluated a multi-branch neuron model to test conductance and geometry inference. The model consists of three branches, each discretized into four cylindrical compartments of radius $2\,\mu m$, for a total of 12 compartments. Each compartment contains standard Hodgkin-Huxley sodium, potassium, and leak channels [6]. Reversal potentials are shared across all branches (Table 2), while conductances and lengths vary by branch (Table 3). The axial resistivity was set to $150\,\Omega \cdot cm$. A current of $2\,nA$ was injected in the first compartment of branch 1 (soma) for $20\,ms$.

With EKF, all the maximum conductances were inferred accurately for all 40 datasets simulated with $\sigma_{obs} = 1\mu V$ (fig. E.5).

With the diagonal EKF, the recovered conductances exhibit a more pronounced bias than in the single-branch case, particularly in branches 2 and 3 (fig. E.6), although the voltage traces remain accurate up to small shifts in these branches (fig. E.7), and the compartment center locations were inferred correctly (fig. E.8). In this multi-branch configuration, the diagonal approximation retains decent accuracy because, despite the relatively dense electrode layout, the long-compartment geometry reduces instances where electrodes are simultaneously close to distinct compartments. Since each compartment only has two neighbors (except at the branching node), this limits off-diagonal measurement coupling. As detailed in Appendix B, this geometry ensures that most of the off-diagonal entries of $(\mathbf{H}^v)^\top \mathbf{H}^v$ remain significantly smaller than terms near the diagonal (fig. E.9). Consequently, the diagonal approximation introduces some bias but continues to provide useful and stable inference.

### E.3 Robustness to Model Misspecification

In the main text, we considered two forms of model misspecification: missing ion channels and incorrect gating kinetics.

|  | Na | K | Leak |
|---|---|---|---|
| Reversal potential (mV) | 53.0 | $-107.0$ | $-88.5188$ |

Table 2: Reversal potentials used in the multi-branch neuron model.

| **Branch** | $g_{Na}$ | $g_K$ | $g_{Leak}$ | Length |
|---|---|---|---|---|
| 1 | 0.12 | 0.02 | 0.0003 | 24.0 |
| 2 | 0.08 | 0.03 | 0.0004 | 50.0 |
| 3 | 0.10 | 0.008 | 0.0001 | 50.0 |

Table 3: Maximum conductances (in S/cm$^2$) and branch lengths (in $\mu$m) in the multi-branch neuron model.

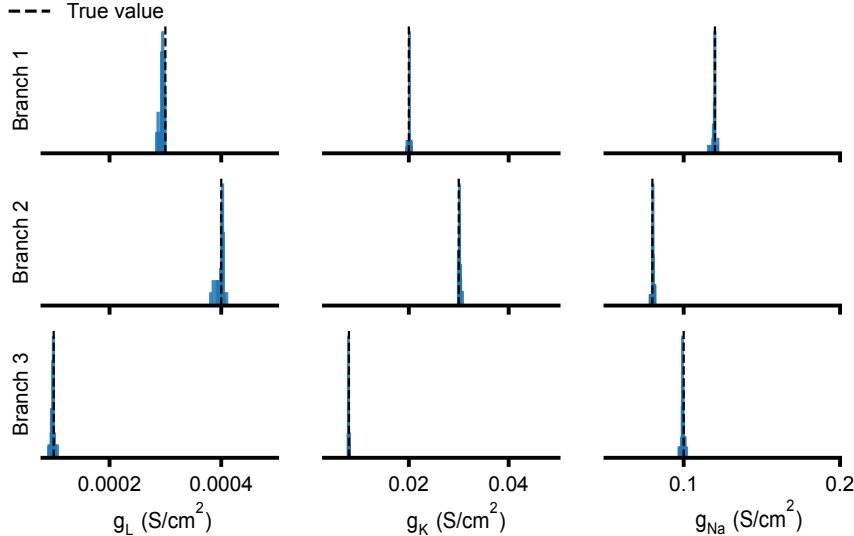

Figure E.5: **Validation of the EKF on a synthetic multi-branch neuron.** The maximum conductances $g_L$, $g_K$ and $g_{Na}$ were inferred accurately for all the branches, across 40 datasets simulated with $\sigma_{obs} = 1\,\mu V$.

**Omitted Ion Channels.** To test robustness to missing channels, we simulated data from a four-compartment branch model (total length $96\mu$m, radius $2\mu$m) containing M-type potassium (KM) and L-type calcium (CaL) channels (Table 5) in addition to Hodgkin-Huxley (HH) channels (Table 4). Inference was performed using a model containing only the HH channels. A constant current was injected for 20 ms. Extracellular voltages were recorded at ten electrodes with a noise level of $\sigma_{obs} = 0.1\,\mu V$. Biophysical parameters for the simulation model are shown in Table 6.

During EKF inference, we fixed the standard deviation of the channel states to $\sigma_{state} = 0.00001\ \mathrm{ms}^{-1/2}$ (same for all the channels) and treated $\sigma_v$ as a free parameter to capture voltage uncertainty due to model mismatch. We performed hyperparameter sweeps over learning rates and initial values of the parameters, selecting the configuration with the highest marginal log-likelihood.

For MSE-based optimization, we used Adam [39] and reported the result with the lowest final MSE.

**Perturbed Gating Dynamics.** To evaluate robustness to incorrect kinetics, we generated data using gating functions from a retinal ganglion cell (RGC) model (Table 8) and performed inference using the original Hodgkin-Huxley kinetics [6]. The neuron was a single cylindrical cable (length $24\,\mu$m, radius $2\,\mu$m), discretized into four compartments. A constant current was injected into the first

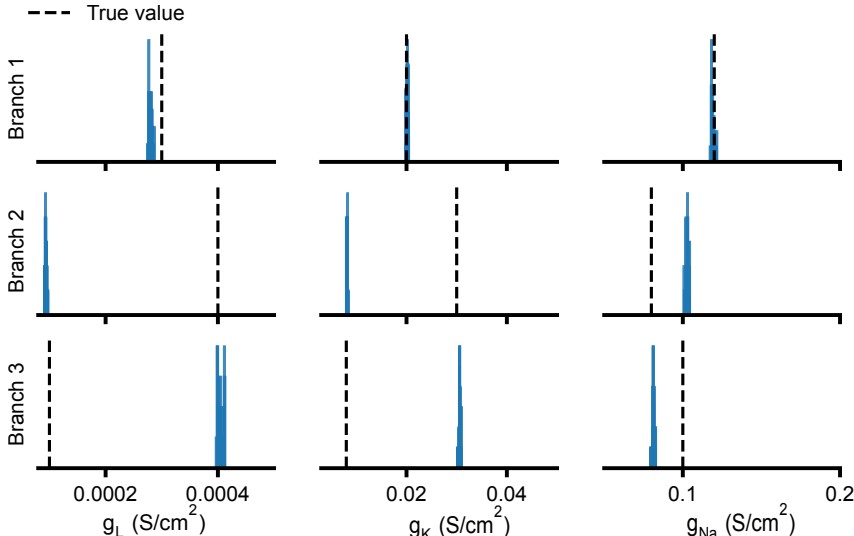

Figure E.6: **Inferred parameters using the diagonal EKF on a synthetic multi-branch neuron.**
We show results across 40 datasets simulated with $\sigma_{\mathrm{obs}} = 1\,\mu\mathrm{V}$. The recovered conductances exhibit a more pronounced bias than in the single-branch case, particularly in branches 2 and 3.

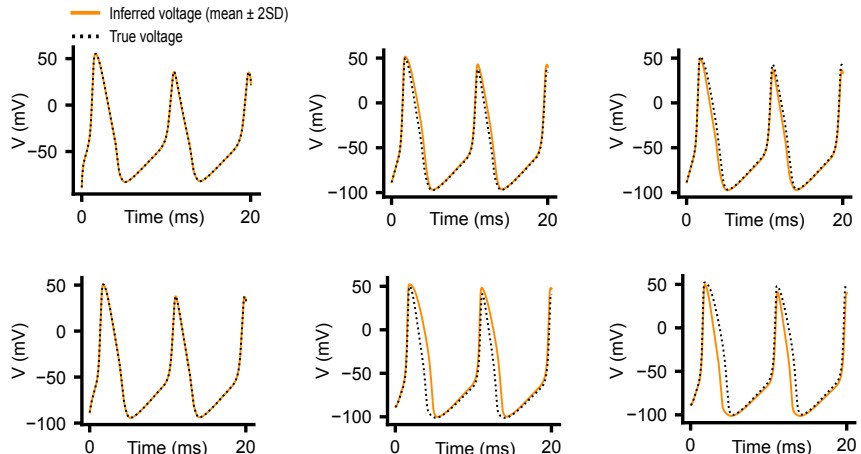

Figure E.7: **Reconstructed voltage for first (top) and last (bottom) compartment of branch 1 (left), 2 (middle) and 3 (right) using the diagonal EKF on a synthetic multi-branch neuron.**
Voltage traces are averaged over 40 randomly simulated datasets with $\sigma_{\mathrm{obs}} = 1\mu\mathrm{V}$. Although conductance estimates exhibit noticeable bias, the reconstructed voltages remain accurate up to small shifts, consistent with a regime in which the diagonal approximation remains adequate.

compartment (soma) for 20 ms. Extracellular voltages were recorded at 10 different locations with noise level $\sigma_{obs} = 0.1\,\mu\mathrm{V}$. Biophysical parameters for the simulation model are shown in Table 7.

During EKF inference, we treated the voltage standard deviation $\sigma_v$ and channel noise levels $\sigma_{\mathrm{K}}, \sigma_{\mathrm{Na}}, \sigma_{\mathrm{Leak}}$ as learnable parameters. While only the sodium and potassium kinetics are misspecified, we allowed separate noise levels for each channel type to provide flexibility in how uncertainty is accounted for. We also used the same $\sigma_v, \sigma_{\mathrm{K}}, \sigma_{\mathrm{Na}}, \sigma_{\mathrm{Leak}}$ across all compartments, though future work could consider compartment-specific noise parameters. As before, we performed grid search over learning rates and initializations, selecting the result with the highest marginal log-likelihood.

For MSE optimization, we used Adam and selected the result with the lowest MSE.

The EKF smoothed posterior mean of the voltage recovers the true voltage trace in each of the four compartments, whereas MSE minimization fails (fig. E.10).

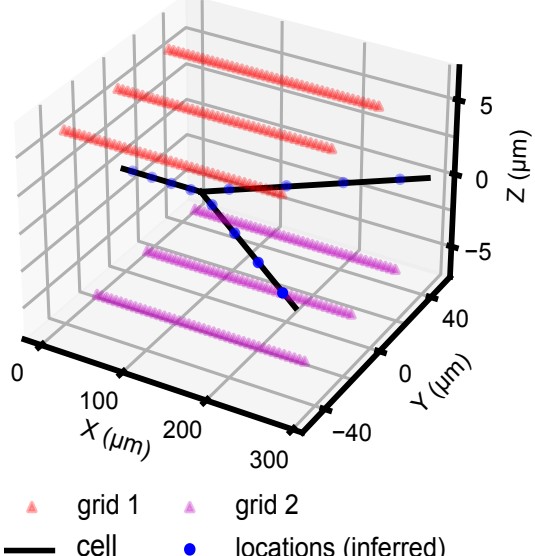

Figure E.8: **Diagonal EKF accurately recovers compartment center locations.** Estimated 3D positions are averaged over 40 randomly simulated datasets with $\sigma_{\text{obs}} = 1\mu\text{V}$. Although conductance estimates exhibit noticeable bias, the reconstructed compartment locations remain accurate, consistent with a regime in which the diagonal approximation remains adequate.

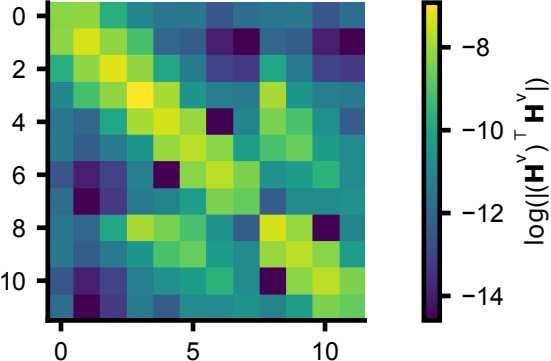

Figure E.9: **Heatmap (in log scale) of $\mathbf{H}^{v\top}\mathbf{H}^v$, the voltage block of $\mathbf{H}^\top\mathbf{H}$.** The largest coefficients occur between adjacent compartments and at branch points, yielding an approximately tridiagonal structure. Off-diagonal entries corresponding to distant compartments are substantially smaller, reflecting the long-compartment morphology, where few electrodes are simultaneously close to two non-neighboring compartments. This structure helps explain why the diagonal approximation remains reasonably effective in this setting.

## F Retinal Ganglion Cell (RGC) Model

We use a morphologically detailed model of a retinal ganglion cell (RGC), based on a reconstruction from Vilkhu et al. [29]. The morphology was originally provided as a NEURON `.hoc` file,[1] which we converted to SWC format for compatibility with JAXLEY. The model includes the full soma and dendritic arbor, and was discretized into 122 compartments, assigning one compartment per branch.

---

[1] https://github.com/ramanvilkhu/rgc_simulation_multielectrode

| Sodium (Na⁺) | Potassium (K⁺) |
|---|---|
| $\alpha_m = \dfrac{0.32(V+47)}{1-e^{-(V+47)/4}}$ | $\alpha_n = \dfrac{0.032(V+45)}{1-e^{-(V+45)/5}}$ |
| $\beta_m = \dfrac{0.28(V+20)}{e^{(V+20)/5}-1}$ | $\beta_n = 0.5\,e^{-(V+50)/40}$ |
| $\alpha_h = 0.128\,e^{-(V+43)/18}$ | |
| $\beta_h = \dfrac{4}{1+e^{-(V+20)/5}}$ | |

Table 4: Rate functions for sodium (Na⁺) and potassium (K⁺) channels used in the single-branch model for the missing channel experiment.

| M-type K⁺ (KM) | L-type Ca²⁺ (CaL) |
|---|---|
| $p_\infty = \dfrac{1}{1+e^{-0.1(V+35)}}$ | $\alpha_q = 0.055\dfrac{V+27}{1-e^{-(V+27)/3.8}}$ |
| $\tau_p = \dfrac{4000}{3.3\,e^{0.05(V+35)}+e^{-0.05(V+35)}}$ | $\beta_q = 0.94\,e^{-(V+75)/17}$ |
| | $\alpha_r = 0.000457\,e^{-(V+13)/50}$ |
| | $\beta_r = \dfrac{0.0065}{1+e^{-(V+15)/28}}$ |

Table 5: Rate functions for M-type potassium (KM) and L-type calcium (CaL) channels used in the single-branch model for the missing channel experiment.

| | Na | K | Leak | KM | CaL |
|---|---|---|---|---|---|
| Maximum conductance (S/cm²) | 0.05 | 0.005 | 0.0001 | 0.000004 | 0.0001 |
| Reversal potential (mV) | 50.0 | $-90.0$ | $-70.0$ | $-90.0$ | 120 |

Table 6: Biophysical parameters for the single-branch neuron model used in the missing channel experiment.

| | Na | K | Leak |
|---|---|---|---|
| Maximum conductance (S/cm²) | 0.12 | 0.02 | 0.003 |
| Reversal potential (mV) | 53.0 | $-107.0$ | $-88.5188$ |

Table 7: Biophysical parameters for the single-branch neuron model used for the kinetic mismatch experiment.

The ion channel model consists of Hodgkin–Huxley (HH) conductances: fast sodium, delayed-rectifier potassium, and leak channels. Voltage-dependent rate constants are listed in Table 8, and channel reversal potentials in Table 9. Region-specific maximum conductances for soma and dendrites are given in Table 10. All values were taken directly from Vilkhu et al. [29], with a minor adjustment to the leak reversal potential to ensure resting membrane potentials near $-65\,$mV across compartments. Note that Vilkhu et al. [29] also consider two additional channels (a calcium channel and a calcium-gated potassium channel as in Fohlmeister et al. [40]), which we neglected in this experiment. Including these channels could be the subject of future work.

The axial resistivity was $143.2\,\Omega \cdot$ cm, the membrane capacitance was $1\,\mu\text{F/cm}^2$ across all compartments, and the extracellular resistivity was $1000\,\Omega \cdot$ cm.

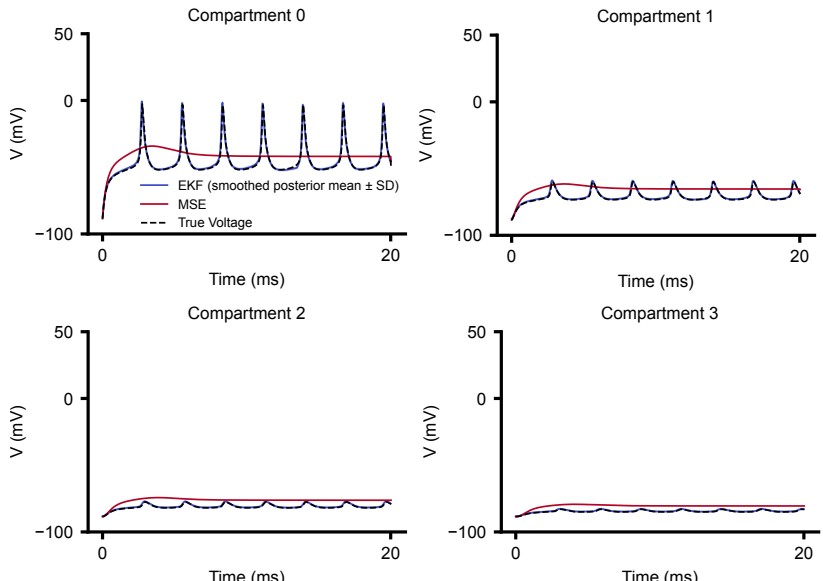

Figure E.10: **Robustness to misspecified channel kinetics.** The EKF smoothed posterior mean of the voltage recovers the true voltage trace in each of the four compartments, whereas MSE minimization fails.

| Sodium (Na$^+$) | | Potassium (K$^+$) |
|---|---|---|
| $\alpha_m = \dfrac{2.725(V + 35)}{1 - e^{-(V+35)/10}}$ | | $\alpha_n = \dfrac{0.09575(V + 37)}{1 - e^{-(V+37)/10}}$ |
| $\beta_m = 90.83\, e^{-(V+60)/20}$ | | $\beta_n = 1.915\, e^{-(V+47)/80}$ |
| $\alpha_h = 1.817\, e^{-(V+52)/20}$ | | |
| $\beta_h = \dfrac{27.25}{1 + e^{-(V+22)/10}}$ | | |

Table 8: Rate functions in the RGC model.

To generate synthetic extracellular recordings, we applied a constant somatic current injection for 10 ms. Voltage recordings were simulated using a synthetic planar multi-electrode array (MEA) with a sampling rate of 40 kHz.

We fit this model using our diagEKF and block-diagEKF implementations. All voltages and channel states were initialized at their steady-state values at rest, and thus we assumed the initial state to be known. We set the initial covariance matrix to $\Sigma_0 = 0.1\mathbf{I}$. Observation noise was also assumed to be known, as commercial devices that record extracellular potentials typically provide noise specifications [16, 38]. Since the state dynamics were well-specified and deterministic, we fixed the standard deviation of the membrane potentials and channel states to small values: $\sigma_v = 0.001\,\mathrm{mV/ms}^{1/2}$, and $\sigma_{\text{state}} = 0.0001\,\mathrm{ms}^{-1/2}$ (same for all the channels) for all compartments. These values ensured numerical stability while avoiding degenerate gradients during optimization.

We optimized conductance parameters using the Adam optimizer. We tried multiple learning rates and initialized values uniformly at random within the bounds shown in Table 11. We ran over 100 random initializations with different seeds and selected the result that achieved the highest marginal log-likelihood.

With the block-diagonal EKF, all maximum conductances were accurately inferred (F.1) and the voltage traces faithfully reconstructed across the cell (F.2).

|                         | Na    | K        | Leak      |
| ----------------------- | ----- | -------- | --------- |
| Reversal potential (mV) | 60.60 | −101.34  | −67.1469  |

Table 9: Reversal potential for ion channels in the RGC model.

|           | $g_{\text{Na}}$ | $g_{\text{K}}$ | $g_{\text{Leak}}$ |
| --------- | ---- | ----- | ------ |
| Dendrites | 0.06 | 0.035 | 0.0001 |
| Soma      | 0.06 | 0.035 | 0.0001 |

Table 10: Maximum conductances in the RGC model (in S/cm$^2$).

| **Region** | $g_{\text{L}}$ | $g_{\text{Na}}$ | $g_{\text{K}}$ |
| -------- | ---------------- | ------------ | ------------- |
| Soma     | [0.0001, 0.005]  | [0.05, 0.1]  | [0.01, 0.05]  |
| Dendrite | [0.0001, 0.005]  | [0.05, 0.1]  | [0.01, 0.05]  |

Table 11: Initialization bounds for the maximum conductances in the RGC model (in S/cm$^2$).

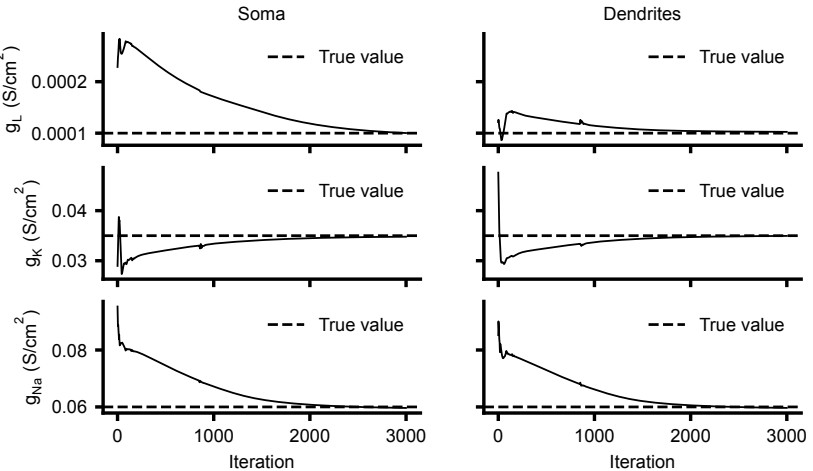

Figure F.1: **Inference of retinal ganglion cell (RGC) properties given multi-electrode array (MEA) recordings (block-diagonal EKF).** Estimation of all the maximum conductances at noise level $\sigma_{\text{obs}} = 0.1\mu\text{V}$.

## G  CA1 Pyramidal Neuron (C10 Model)

We use the C10 model of a CA1 pyramidal neuron, originally developed by Cutsuridis et al. [32]. The morphology is provided as a NEURON .hoc file,[2] which we converted to SWC format for compatibility with JAXLEY. This file specifies the full 3D morphology, including the diameters and lengths of all the branches (Table 14). As shown in fig. G.1, the neuron is divided into 15 anatomical sections.

Following Cutsuridis et al. [32], we applied the compartmentalization heuristic of Carnevale and Hines [7] to determine the number of compartments per section. Specifically, a section of length $L$ is discretized into

$$N = \left\lfloor \frac{1}{2} \left( \frac{L}{0.1 \cdot \lambda_f(100)} + 0.9 \right) \right\rfloor \cdot 2 + 1,$$

[2] https://github.com/tomko-neuron/HippoUnit/tree/master/Cutsuridis_CA1_2010

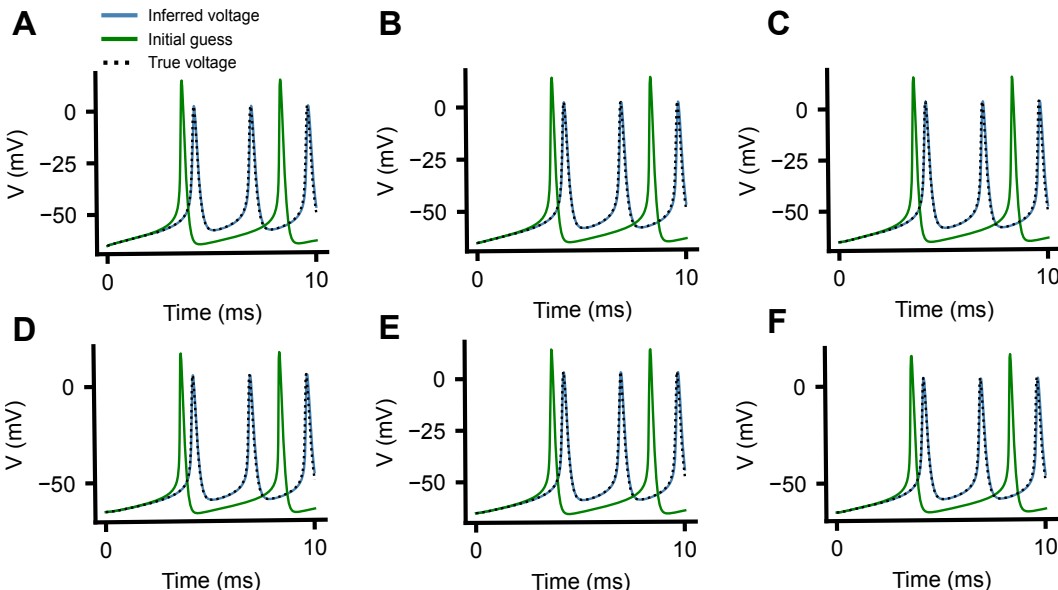

Figure F.2: **Membrane voltage recovery for the RGC model in six compartments (bloc-diagonal EKF). A**: soma, **B**: compartment 31, **C**: compartment 61, **D**: compartment 91, **E**: compartment 106, **F**: compartment 122. The ordering of the compartments is determined by the SWC file.

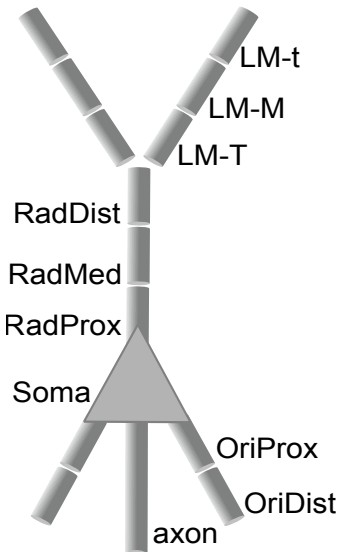

Figure G.1: **Morphology of the C10 model.** LM-t: lacunosum moleculare (thin), LM-M: lacunosum moleculare (medium), LM-T: lacunosum moleculare (thick), RadDist: radiatum distal, RadMed: radiatum medial, RadProx: radiatum proximal, OriProx: oriens proximal, OriDist: oriens distal.

where the frequency-dependent electrotonic length is defined as $\lambda_f(f) = \frac{1}{2}\sqrt{\frac{d}{\pi f R_a C_m}}$, with $d$ the diameter, $R_a$ the axial resistivity, and $C_m$ the specific membrane capacitance. This yields a total of 60 compartments.

Each compartment contains Hodgkin–Huxley (HH) conductances: sodium, potassium, and leak. The sodium current, however, uses a three-gate formulation with gating vector $\lambda_{\text{Na}} = (m, h, s) \in [0, 1]^3$. The total transmembrane current released by a compartment is given by:

$$I_m = g_{\text{Na}}\, m^2 h s (V - E_{\text{Na}}) + g_{\text{K}}\, n^2 (V - E_{\text{K}}) + g_{\text{L}}(V - E_{\text{L}}). \tag{15}$$

| Location | $m$ | $h$ | $n$ |
|---|---|---|---|
| Soma/Axon | $m_\infty = \dfrac{1}{1+e^{\frac{V+44}{-3}}}$ | $h_\infty = \dfrac{1}{1+e^{\frac{V+49}{3.5}}}$ | $n_\infty = \dfrac{1}{1+e^{\frac{V+46.3}{-3}}}$ |
| | $\tau_m = 0.05$ ms | $\tau_h = 1.0$ ms | $\tau_n = 3.5$ ms |
| Dendrite | $m_\infty = \dfrac{1}{1+e^{\frac{V+40}{-3}}}$ | $h_\infty = \dfrac{1}{1+e^{\frac{V+45}{3}}}$ | $n_\infty = \dfrac{1}{1+e^{\frac{V+42}{-2}}}$ |
| | $\tau_m = 0.05$ ms | $\tau_h = 0.5$ ms | $\tau_n = 2.2$ ms |

Table 12: Steady-state functions and time constants for gates $m$, $h$, and $n$ in the C10 model.

| s | Formula |
|---|---|
| $s_\infty(V)$ | $\dfrac{1 + \text{Na\_att} \cdot e^{(V+60)/2}}{1 + e^{(V+60)/2}}$ |
| $\tau_s(V)$ | $\max\left( \dfrac{e^{0.0854(V+60)}}{0.0003(1 + e^{0.427(V+60)})},\ 3.0 \right)$ |

Table 13: Kinetics of sodium attenuation gate $s$ as a function of attenuation factor Na_att in the C10 model.

| Section | Diameter (μm) | Length (μm) | Na_att |
|---|---|---|---|
| Soma | 10 | 10 | 0.8 |
| Axon | 1 | 150 | 1.0 |
| RadProx | 4 | 100 | 0.5 |
| RadMed | 3 | 100 | 0.5 |
| RadDist | 2 | 200 | 0.5 |
| OriProx | 2 | 100 | 1.0 |
| OriDist | 1.5 | 200 | 1.0 |
| LM-T | 2 | 100 | 0.5 |
| LM-M | 1.5 | 100 | 0.5 |
| LM-t | 1 | 50 | 0.5 |

Table 14: Detailed morphology of the C10 model and sodium channel attenuation (`Na_att`) per section.

Moreover, unlike the canonical HH model, the gating kinetics vary across anatomical regions [41]. Gating variables evolve according to first-order kinetics:

$$\tau_\lambda(V)\frac{d\lambda}{dt} = \lambda_\infty(V) - \lambda.$$

The steady-state functions $\lambda_\infty(V)$ and time constants $\tau_\lambda(V)$ for $m$, $h$, and $n$ vary by region and are listed in Table 12.

The third gating variable – $s$ – models sodium attenuation. Its dynamics are consistent across regions except for a region-specific attenuation factor $\text{Na\_att} \in [0, 1]$. The expressions for $s_\infty(V)$ and $\tau_s(V)$ at fixed Na_att are given at temperature $T = 34°C$ (Table 13).

The attenuation factor `Na_att` is defined per section based on Migliore et al. [35] (Table 14).

Reversal potentials follow Cutsuridis et al. [32], with a small adjustment to $E_{\text{L}}$ to enforce a resting potential near $-70$ mV across compartments (Table 15).

The maximum conductances for each region were taken directly from Cutsuridis et al. [32] (Table 16).

We used the following physiological constants: axial resistivity $R_a = 150\ \Omega \cdot \text{cm}$, specific membrane capacitance $C_m = 1\ \mu\text{F}/\text{cm}^2$, and extracellular resistivity $\rho_{\text{ext}} = 300\ \Omega \cdot \text{cm}$. While we assume homogeneous extracellular resistivity, this simplification neglects known spatial variability [25],

|  | Na | K | Leak |
|---|---|---|---|
| Reversal potential (mV) | 50.0 | $-80.0$ | $-123.0$ |

Table 15: Reversal potential in the C10 model.

| Region | $g_L$ | $g_{Na}$ | $g_K$ |
|---|---|---|---|
| Soma | 0.0002 | 0.007 | 0.0014 |
| Axon | 0.000005 | 0.1 | 0.02 |
| OriProx | 0.000005 | 0.007 | 0.000868 |
| OriDist | 0.000005 | 0.007 | 0.000868 |
| RadProx | 0.000005 | 0.007 | 0.000868 |
| RadMed | 0.000005 | 0.007 | 0.000868 |
| RadDist | 0.000005 | 0.007 | 0.000868 |
| LM | 0.000005 | 0.007 | 0.000868 |

Table 16: Maximum conductances in the C10 model (in S/cm$^2$).

| Region | $g_L$ | $g_{Na}$ | $g_K$ |
|---|---|---|---|
| Soma | [1e-4, 4e-4] | [0.003, 0.015] | [0.001, 0.005] |
| Axon | [1e-6, 1e-5] | [0.08, 0.12] | [0.015, 0.03] |
| Dendrite | [1e-6, 1e-5] | [0.003, 0.015] | [0.0005, 0.002] |

Table 17: Initialization bounds for the maximum conductances in the C10 model (in S/cm$^2$).

which could be addressed in future work. Note that Cutsuridis et al. [32] also consider additional channels, which we neglected here. Including them in our model could also be the subject of future experiments.

Synthetic extracellular recordings were generated by injecting a constant current into the soma for 30 ms. Voltage traces were simulated using a synthetic Neuropixel Ultra probe at a 40 kHz sampling rate, with noise standard deviation $\sigma_{obs} = 0.1\mu V$.

The true location of the cell was generated by applying a rigid-body transformation—consisting of a 3D rotation and translation—to the positions of the compartment centers defined via the SWC file. The geometric parameters thus included three rotation angles (one for each axis), applied to the base positions of the compartment centers obtained from the SWC file, and a 3D translation vector applied to the base position of the soma compartment.

We fit this model using the block-diagonal EKF. All voltages and channel states were initialized at their steady-state values at rest, and thus we assumed the initial state to be known. We set the initial covariance matrix to $\Sigma_0 = 0.1\mathbf{I}$. Observation noise was also assumed to be known, as commercial devices that record extracellular potentials typically provide noise specifications [16, 38]. Since the state dynamics were well-specified and deterministic, we fixed the standard deviation of the membrane potentials and channel states to small values: $\sigma_v = 0.001\,\mathrm{mV/ms^{1/2}}$, and $\sigma_{\mathrm{state}} = 0.0001\,\mathrm{ms^{-1/2}}$ (same for all the channels) for all compartments. These values ensured numerical stability while avoiding degenerate gradients during optimization.

We optimized maximum conductances and geometric parameters using the Adam optimizer. We tried multiple learning rates and initialized the maximum conductances uniformly at random within the bounds shown in Table 17. The rotation angles were initialized uniformly on $[0, 2\pi)$, and translations were initialized to place the soma near its true location. We chose this initialization to reduce the risk of converging to a mirror-reflected solution, which produces identical extracellular signals when using a planar probe (see Section D). We ran 100 random initializations with different seeds and selected the result that achieved the highest marginal log-likelihood.

Although the marginal log-likelihood converged to its true value after a few hundred iterations (fig. G.3F), and both the voltage traces (fig. G.3A–E) and compartment positions (fig. 5A) were

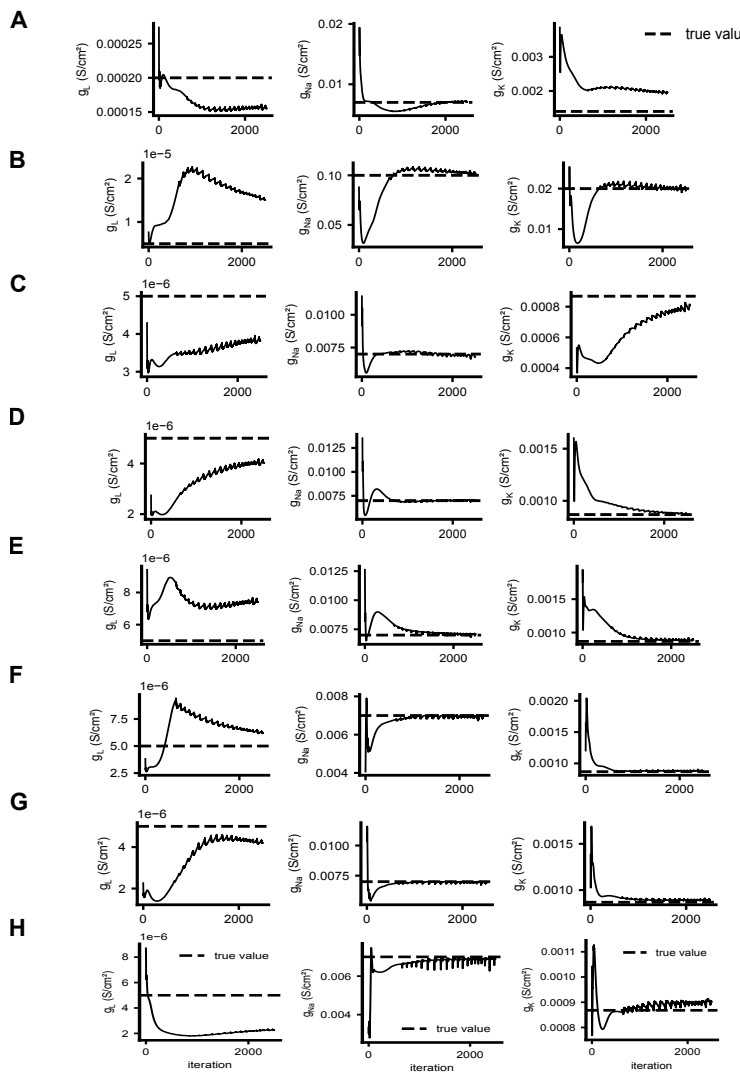

Figure G.2: **Inference of the maximum conductances of the C10 model in all the regions. A**: Soma, **B**: Axon, **C**: OriProx, **D**: OriDist, **E**: RadProx, **F**: RadMed, **G**: RadDist, **H**: LM. While most maximum conductances converged to their true values, many–especially the leak conductances–were still fluctuating.

accurately recovered across the cell, the leak conductances did not converge to their true values or even continued to fluctuate (fig. G.2) —reflecting the well-known degeneracy in conductance-based neuron models, where multiple parameter combinations can produce nearly identical voltage dynamics.

## H   DiagEKF vs. EKF runtime

We compare the computational efficiency of the standard Extended Kalman Filter (EKF) and our sparse diagonal approximation (DiagEKF). To isolate performance differences, we consider a simplified neuron model: a $24\,\mu$m cable equipped with standard Hodgkin–Huxley channels [6]. A constant current is injected for 20 ms, and extracellular voltage is recorded at a single electrode at a 40 kHz sampling rate.

The neuron is discretized into $N$ compartments, with $N \in \{1, 10, 20, 50, 100, 200\}$. As $N$ increases, the number of state variables (membrane voltages and gating variables) grows proportionally. Importantly, the gating dynamics in these models are highly local: each variable typically depends only on

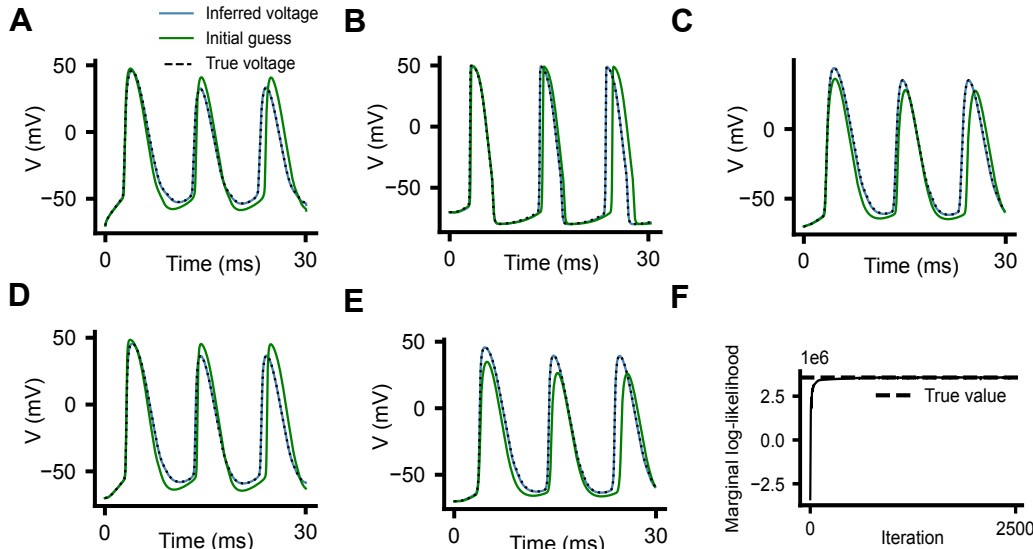

Figure G.3: **Recovery of the membrane voltage in different regions of the pyramidal cell. A**: Soma, **B**: Axon, **C**: RadDist, **D**: OriDist, **E**: LM. **F**: Marginal log-likelihood trace during optimization. Despite some maximum conductances failing to converge to their true values, the membrane voltage was recovered accurately across the cell and the marginal log-likelihood converged to its true value, reflecting the well-known degeneracy in conductance-based neuron models.

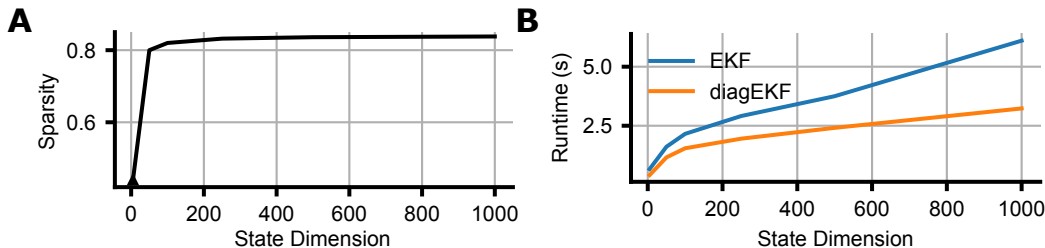

Figure H.1: **Computational efficiency of our diagonal approximation to the EKF filtering distribution A**: Sparsity of the Jacobian of the dynamics function over state dimension. Especially large biophysical models are sparse, suggesting that sparse approximations to the filtering covariance could provide accurate but efficient inference. **B**: Average runtime over $500$ updates for the EKF and sparse diagEKF versus state dimension for a $20ms$ time horizon. The EKF with diagonal approximation is twice as fast as EKF with a dense covariance for large models.

its own previous value and the voltage of its corresponding compartment. As a result, the Jacobian of the dynamics function quickly becomes very sparse when the number of states increases (fig. H.1A).

To evaluate performance, we ran both EKF and DiagEKF for 500 iterations on an NVIDIA H100 GPU for each value of $N$. As shown in fig. H.1B, DiagEKF achieves approximately 2× speedup over the dense EKF in high-dimensional settings. This highlights the computational advantage of our diagonal approximation in large biophysical neuron models.

