# OpenReview forum: "Identifying multi-compartment Hodgkin-Huxley models with high-density extracellular voltage recordings"
_NeurIPS.cc/2025/Conference — NeurIPS 2025 poster_

### Official Review · Reviewer_tGNX · 2025-06-05

**Clarity:** 3
**Significance:** 2
**Originality:** 3
**Rating:** 4
**Confidence:** 3

**Summary:**

The paper introduces a novel approach to estimating the HH parameters of neurons in high density arrays. Authors show good performance on simulated datasets + two neurons from real-wolrd datasets (one 2D retinal ganglion cell and one CA1 hippocampal pyramidal cell).

**Questions:**

This is a very nice paper, but I believe the clarity of the paper could be improved, which would make the paper a bit more accessible. I struggled a bit understanding the following points:
 - I would mention that the method is only evaluated / used for in vitro recordings in the intro, as this is only clear after reading section 5.
 - Similarly, why contrast your approach with spike sorting if you're using the output of spike sorting (lines 34-36)? Especially as the method is only evaluated on in-vitro data, I believe it could in the future be complementary to spike sorting, and these lines make it seem like you're targetting in vivo recordings (I'm not sure if it's needed to criticize an approach designed to tackle a very different problem, o reven mention spike sorting)
 - What do the "states" (channel states) represent?
 - The parameters of the HH evolve in time (or depend on the neural activity, and events such as bursting, or other recording artefacts that might hinder the recorded signal). Are you restricting the approach to well-defined neurons in specific timesteps? Could you show the evolution of the parameters in time? Or maybe they are stable in-vitro?
 - MEA arrays are not as high resolution as NP Ultra. The motivation mentions NP Ultra but if the method works well on MEA, it should work well on NP1/NP2 as well. Could you extend your analysis to these probes as well? (if there's available data)
 - You're also not citing the more up-to-date paper, not sure if you should update it to the more recent citation? (Ultra-high density electrodes improve detection, yield, and cell type identification in neuronal recordings, Ye et al., 2024)

**Ethical Concerns:**

["NO or VERY MINOR ethics concerns only"]

**Final Justification:**

The paper proposes a cool novel model for HH model parameter estimation in high density electrodes, but in my opinion the evaluation is a bit limited (few cells + lack of comparison to other methods or justification of design choices). Thus, I recomend weak accept

**Limitations:**

Limitations are well adressed but I would clarify the scope of the paper in the intro - to make it clear to the reader that the method is so far only evaluated and used on in vitro data.

**Paper Formatting Concerns:**

No formatting concerns

**Quality:**

3

**Strengths And Weaknesses:**

Strengths: It is a very interesting paper with a novel model for HH parameter estimation, and very good results.

Weaknesses:

I find the paper to be very good overall but the main weakness lies in the lack of comparison to existing methods.
other methods have been proposed to estimate the parameters of a HH model. For example Multi-objective Differential Evolution (DEMO, Lois et al., 2022) and ElectroPhysiomeGAN (Kim et al., 2025). These methods might not have been developed and tested on high-density extracellular voltage recordings but it would be worth adding some comparisons to one of these methods + citing these approaches and explaining the differences with your approach.

The results could be improved as well by showing results on more than two cells. Why not extend it to many more cells and report statistics (overall performance of true vs reconstructed signals). It would be great to show this to make sure these two cells are not cherry picked examples. Are you ilmited by data availability (as these are in vitro recordings)?

---

> ### Author Rebuttal · Authors · 2025-07-31
>
> We thank the reviewer for taking the time to review our submission and for their thoughtful and constructive feedback. We were especially encouraged by the reviewer’s positive comments on the novelty of our work. Below, we address each of the six points made by the reviewer and hope to provide the necessary clarifications:
>
> - The reviewer suggests clarifying earlier that the method is not applied to in vivo data. We fully agree and thank the reviewer for this suggestion. Although extracellular voltage recordings are well-suited for in vivo acquisition, the current study does not yet include any in vivo experiments. We will clarify this explicitly in the introduction.
>
> - The reviewer suggests clarifying how our method relates to spike sorting. We thank you for raising this important point. Our intention in lines 34–36 was simply to note that extracellular recordings are typically used to obtain spike times and putative neuron labels via spike sorting, but the spike waveform contains rich information that can be used for fitting more detailed biophysical models as well. However, we agree that the current wording could be misleading since our method also relies on spike sorting to identify EAPs. We will revise the introduction to make this distinction clearer.
>
> - The reviewer asks for clarification on what is meant by "channel states". In Hodgkin–Huxley models, each ion channel $c\in\mathcal{C}$ (where $\mathcal{C}$ is the set of all ion‑channel types included in our model) is governed by one or more gating variables whose values lie in [0,1]. For example, the fast Na+ current has activation and inactivation gates $m(t)$ and $h(t)$, whereas the delayed‑rectifier K+ current has a single activation gate $ n(t) $. We collect these variables in a vector $\lambda_c(t)  =  (m_c(t),h_c(t),… )$, which represents the probability that the molecular gates of channel $ c $ are open at time $t$. We define the channel states at time $t$ as the concatenation of all such vectors over channel types $c\in\mathcal{C}$ and also compartments. We will revise the manuscript to make this definition—and its connection to the underlying biophysics—more explicit.
>
> - The reviewer asks whether the Hodgkin–Huxley parameters are constant or time-varying, and whether the model can handle dynamic changes in parameters. The Hodgkin-Huxley parameters we infer are the maximum conductances $g_c$​ of each channel type $c$. These biophysical constants are assumed to remain fixed throughout the length of the recordings in each of our experiments. What varies in time are the effective conductances defined as the maximum conductance times the proportion of ion channels opened at time $t$ (a number between 0 and 1).
> Although not considered in our current work, one possible source of time-varying conductances is short-term plasticity or activity-dependent modulation. Such effects could be incorporated by introducing additional differential equations into the dynamics model—governing, for instance, how the maximum conductances $g_c$ evolve over time. These added components would introduce new latent states and parameters, which could also be inferred within our framework by extending the EKF and continuing to maximize the marginal likelihood.
>
> - The reviewer suggests it would be more appropriate to emphasize applicability to a broader range of probes, including NP1 and NP2. We fully agree and appreciate this suggestion. Although the Neuropixels Ultra study initially motivated our focus on dense extracellular recordings, we agree that our method should also apply to NP1 and NP2. Indeed, NP1 and NP2 have pitches of 20 µm and 15 µm, respectively, while the MEA used in our RGC experiments has a 30 µm spacing. We will revise our framing to avoid focusing so specifically on NP Ultra and instead emphasize that the method is compatible with a range of high-density probes. We have not yet run experiments on NP1/NP2 data, but this would be a natural direction for future work.
>
> - The reviewer notes that a more recent version of the Neuropixels Ultra paper is now available and should be cited. We thank the reviewer for catching this mistake. We will replace it with the correct reference to Ye et al., 2024, in the camera‑ready version.

---

> > ### Comment · Reviewer_tGNX · 2025-08-01
> >
> > I thank the authors for their response to my reviews.
> >
> > They clarified the questions I had and hope that these clarifications can be added to the manuscript to improve its clarity.
> >
> > As authors have not addressed the weaknesses I outlined (comparison to other methods + results on more cells), I will not increase my score.

---

> > > ### Author Response · Authors · 2025-08-02
> > >
> > > We thank the reviewer for their follow-up and apologize for addressing only the questions and not the "Weaknesses" section in our initial response.
> > >
> > > - We acknowledge the reviewer’s concern regarding comparisons to other methods. To our knowledge, no existing method has been applied to fitting Hodgkin–Huxley models directly from high-density extracellular recordings, which makes benchmarking challenging. We thank the reviewer for highlighting DEMO and ElectroPhysiomeGAN. Extending these approaches to the extracellular setting would indeed be very interesting. However, this would not be straightforward, as it would require adapting their architectures to incorporate a model of extracellular recordings (similar to the one used in our work), as well as further modifications to handle model misspecification in the way our method does. These are valuable directions for future work, and we appreciate the reviewer for pointing out these papers. We will cite both methods in the revised introduction when discussing related work on fitting HH models from intracellular data.
> > >
> > > - We agree with the reviewer that evaluating the method on more cells would be valuable. We would like to note that we first validated the method on synthetic morphologies, and for each experiment in that setting, we assessed performance across dozens of simulated datasets. These consistently yielded accurate recovery of voltages and parameters, giving us confidence in the method’s robustness. That said, we agree that extending to more real cells would further strengthen the work. We are currently working on applying our method to additional recordings from different retinal ganglion cells with plans to report broader results in future work.

---

> ### Author Response · Authors · 2025-08-09
>
> Thank you again for your thoughtful comments on our paper. We wanted to know whether you had a chance to read our follow-up response addressing the “Weaknesses” section, and if it helped address your concerns.
> We appreciate your time and feedback.

---

### Official Review · Reviewer_aSRG · 2025-07-01

**Clarity:** 3
**Significance:** 2
**Originality:** 3
**Rating:** 4
**Confidence:** 3

**Summary:**

This paper discusses a new method for estimating the electrical activity of neurons using extracellular voltage measurements. Traditionally, biophysical models like the Hodgkin-Huxley model require intracellular in vitro voltage data to estimate key parameters. With the advent of high-density probes that capture extracellular voltage from multiple sites around a neuron, the authors propose a technique to indirectly measure the neuron’s membrane voltage, biophysical properties, and its position relative to the probe.

The method leverages an extended Kalman filter (EKF) to infer membrane voltage and ion channel states, using efficient, differentiable simulators for computation. It also employs gradient-based optimization to learn the model parameters by maximizing the marginal likelihood. The approach is validated with both simulated data and real neuron morphologies, showing its potential for more accurate and practical neuron modeling using extracellular recordings alone.

**Questions:**

1. Why was the Hodgkin-Huxley model chosen for this method, especially given the assumptions about the gating variables and model dimensions, and could an abstract non-linear model offer more flexibility or accuracy in capturing neuron behaviors?

2. What strategies or modifications are in place to handle neurons that don't fit the traditional Hodgkin-Huxley model, especially those with unconventional ion channels or non-linear dynamics? For example, neuros with heterogeneous spike patters, like regular and fast spiking, and chattering, adaption, etc.

3. How does the method address uncertainties related to the number of gating variables in the Hodgkin-Huxley model, and what is the impact of fixing these dimensions on the model's flexibility?

4. Can you elaborate on the potential limitations or inaccuracies introduced by the EKF’s linearization process, and how might these affect the quality of parameter estimates, particularly in real-time applications?

5. What steps are being taken to improve the method's scalability to larger networks of neurons, and how are multi-compartment interactions and signal overlap handled in the current framework, particularly with regard to sensor density, the number of compartments, and the interconnection terms between compartments?

**Ethical Concerns:**

["NO or VERY MINOR ethics concerns only"]

**Final Justification:**

I think the authors have not been able address the critical concern regarding the question on scalability. Even a single neuron are sometimes decomposed into multiple compartments and they are modeled and analyzed. My question on scalability was on such multi-compartment models, not necessarily multiple neurons. Other assumptions on gating variables and their numbers are significantly limiting the applicability of the method. Hence, I do not intend to change my rating.

**Limitations:**

1. Please include all the assumptions that are required to be satisfied for the algorithm to work.
2. Motivation to use HH-based model and the assumptions made regarding the inference need stronger argument. Otherwise, the method, while interesting, may not be useful.

**Quality:**

2

**Strengths And Weaknesses:**

The method is non-invasive, relying solely on extracellular recordings, which makes it practical for in vivo studies.

It uses an Extended Kalman Filter (EKF), which allows for real-time estimation of membrane potentials and ion channel states.

Validation with simulated and real neuron morphologies shows its potential for accurate modeling of neurons.

It also accounts for the complex morphology of neurons, which is crucial for realistic simulations.

The method depends heavily on the quality of extracellular data, which is often noisy and may not fully capture the necessary details of individual neuron activity, particularly in densely packed regions.

Several assumptions are needed for the proposed method to be effective. These are not explicitly stated.

The method assumes that the Hodgkin-Huxley model is appropriate for all neuron types, which might not be true for more complex or unconventional neuron behaviors.

Number of channel variables are assumed to be known, this fixes dimensions for the inferred model. In other words, prior knowledge of the number of gating variables should be known. This assumption makes the inference in the form of HH model to be unnecessary. In case this assumption cannot be met, then any abstract dynamic model can be used to do the inference instead of HH model structure.

The use of EKF and gradient-based optimization also introduces computational complexity, which could be a bottleneck in large-scale systems or real-time applications. The linearization along with the gradient-based updates may introduce suboptimal estimates and may require further analysis to characterize uncertainties in the estimates.

Estimating the neuron's position relative to the probe adds another layer of complexity, and any errors in this estimation could lead to inaccuracies. Validation with simulated data might not capture all the noise and artifacts found in real-world recordings.

The number of compartments and the interconnection term in the model needs further analysis as they are dependent on the density of the sensor array.

Lastly, while the method is suited for single-neuron modeling, its application to larger networks could become more challenging due to the complexities of multi-neuron interactions and signal overlap.

---

> ### Author Rebuttal · Authors · 2025-07-31
>
> We thank the reviewer for taking the time to review our submission and highlighting some interesting clarification points. Below we address each of the five questions raised and hope our responses provide the necessary clarifications:
>
> - The reviewer asks about the motivation for choosing the Hodgkin–Huxley framework and its ability to capture neuron behaviors flexibly. We chose the Hodgkin–Huxley (HH) framework over an abstract model because our aim extends beyond reproducing voltage traces—we want to recover biophysically meaningful parameters such as maximal conductances and the neuron’s spatial relationship to the probe, quantities of interest to neuroscientists. HH models are also highly extensible: virtually any ion channel found in the literature can be incorporated by specifying its kinetics. The real difficulty lies in fitting such richly parameterized models, not in their expressiveness. We acknowledge, however, that HH models presuppose some prior knowledge of which channels are present, and that uncertainty can limit identifiability; our method, therefore, includes mechanisms (process noise) to remain robust when the true channel set is only partially known.
>
> - The reviewer raises a concern about how the method generalizes to neurons with unconventional ion channels or spiking behaviors. Our framework starts from the canonical view of a neuron as a discretized cable: each compartment is an RC circuit equipped with voltage‑gated or calcium-gated conductances. This multi‑compartment Hodgkin–Huxley (HH) formulation has been at the center of cellular neurophysiology for decades precisely because it is mechanistic, extensible, and cell‑type agnostic—you obtain different electrophysiological phenotypes by choosing the appropriate set of channels and kinetics. In practice, the HH framework comfortably accommodates the diversity of spiking behaviors: spike‑frequency adaptation (A-current, Ca-activated K+), bursting/chattering (calcium channels combined with Ca-dependent K+ currents), fast spiking (Kv3-family K+ channels).
> Our current experiments focused on the standard Na/K/Leak HH channels to make the methodological contributions clear, but the inference machinery is unchanged when richer channel repertoires are introduced: adding a channel simply augments the state vector and parameter set, and the EKF (or its diagonal variant) and gradient-based learning proceed identically. In short, the limitation is not the HH formalism but which channels you choose to include—a choice that can be guided by prior anatomical/physiological knowledge for each neuron class.
>
> - The reviewer asks about how the method addresses uncertainty or misspecification in the set of gating variables and the implications of fixing the model’s structure in advance. While the neuron’s type and location provide useful prior knowledge about which ion channels are present, we agree with the reviewer that misspecification is almost inevitable because (1) ion channels might have been neglected, (2) gating kinetics might be approximate. Indeed, the standard Hodgkin-Huxley rate functions are empirical fits from voltage-clamp data and may not capture all the biophysical subtleties. Real gating variables often deviate from these idealized kinetics due to modulatory factors such as temperature.
> We show that naively fitting HH models with mean-squared error (MSE) struggles under these mismatches—it fails to recover the true membrane voltage and may even miss the correct spike count. To address this, we model both voltage and gating variables as stochastic processes, adding small white-noise terms at each time step. This approach, combined with an Extended Kalman Filter (EKF), allows the model to tolerate structural errors and still accurately track the true voltage, even in cases of channel misspecification (cf. Fig. 3).
>
> - The reviewer seeks clarification on the impact of EKF linearization. We demonstrated that MSE minimization and the EKF yield similar, unbiased parameter distributions (cf. Fig. 2c). A similar lack of bias was also observed in the multicompartment setting (e.g., the 3-branch model). This supports the idea that linearization provides a robust approximation of the marginal likelihood and introduces only negligible bias in parameter estimation. Although previous work has shown that the EKF can become unstable at low sampling rates (Lankarany et al., 2014), we did not encounter this issue as we focused on high-resolution recordings such as those from Neuropixels or MEAs.
>
> - The reviewer asks about the scalability of the method to larger neuronal structures or networks, and how the model handles inter-compartmental interactions. While our study focused on single neurons, leaving full network inference for future work, we are already able to handle large single cells efficiently by two design choices: (i) diagonal-covariance EKF—we keep only the diagonal of the filtering covariance, and (ii) sparse Jacobians—each gating variable depends only on its own previous value and the local voltage, so the dynamics’ Jacobian is sparse. These choices make every EKF update run in time proportional to the state dimension, allowing us to fit morphologies as large as the 122-compartment retinal ganglion cell. Inter-compartment interactions enter solely through axial currents (set by the cable’s axial resistivity), and those currents feed directly into the Hodgkin–Huxley equations, so their effect on the transmembrane currents—and hence on the probe’s extracellular signal—is captured automatically within the same formalism.

---

### Official Review · Reviewer_ntGU · 2025-07-12

**Clarity:** 4
**Significance:** 3
**Originality:** 3
**Rating:** 5
**Confidence:** 4

**Summary:**

This paper addresses the estimation of the parameters of a multi-compartment Hodgkin Huxley model with extracellular voltage recordings. The authors use an Extended Kalman Filter to infer the membrane potential, channel states, and the biophysical parameters of a neuron with simple and complex morphologies. This is a fairly novel and interesting study for both the methodology and the application.

**Questions:**

See above weaknesses.

**Ethical Concerns:**

["NO or VERY MINOR ethics concerns only"]

**Limitations:**

Yes

**Quality:**

4

**Strengths And Weaknesses:**

Strengths:
- This is a novel application of the SSM methods to reconstruct complex HH parameters.
- The simulations include simple and complex multi-compartment morphologies, and also contain different noise levels as well as channel misspecification results (channels omitted or gating dynamics altered).
- The simulations also included results on real neuron morphologies, namely a 2D retinal ganglion cell and CA1 hippocampal cell.
- The formulation of the SSM and scaling of the EKF, as well as the use of SPARSEJAC, are novel here.

Weaknesses:
- Is the diagonal Gaussian assumption in the EKF a reasonable approximation in this context? Detail how valid this is, if possible quantitatively.
- The application of the methods on real Neuropixels or MEA data would strengthen the manuscript.
-  Can the authors address the issue of parameter identifiability a bit more in detail, especially in simulation? Would it be possible to obtain the identifiable subspace, for example, for the larger models that are considered?
- It would be good to compare the methods to alternative inference methods apart from MSE minimization.

---

> ### Author Rebuttal · Authors · 2025-07-31
>
> We thank the reviewer for taking the time to review our submission and for their positive comments! We were particularly pleased by the comments on the novelty of our work. Below, we address each of the four points made in the "weaknesses" section of the review and hope to provide the necessary clarifications:
>
> - The reviewer raises an important question about the validity of the diagonal approximation used in the EKF. We adopted the diagonal approximation to the EKF covariance primarily to reduce memory and computational cost. While this simplification is not biophysically exact, it is guided by observed structure in the filtering covariance matrices obtained during full EKF runs. In multi-compartment models, voltage updates depend on neighboring compartments, and channel states are tightly coupled to local voltage. This suggests that latent variables within the same compartment—and to a lesser extent, across adjacent compartments—should be more correlated, leading to filtering covariances with an approximate banded structure.
> To investigate this, we ran the full EKF on synthetic data (with known parameters) and examined the resulting filtering covariance matrices over time. We observed that, in many time steps, these covariances exhibited an approximate banded structure, with the largest values often concentrated on the main diagonal. This observation motivated us to start with a diagonal approximation as a computationally efficient baseline that captures the dominant structure.
> Empirically, we found that the diagonal EKF performs very well in Hodgkin–Huxley models, including large multi-compartment neurons. In particular, we validated it on the retinal ganglion cell (RGC) model with over 600 latent dimensions, where it produced accurate voltage reconstructions and reliable conductance estimates. We also tested it on synthetic morphologies such as the 3-branch, 12-compartment neuron. Although the full EKF was used in the paper for this experiment (since it was tractable at that scale), we verified that the diagonal approximation gave nearly identical results: accurate recovery of membrane voltage traces and compartment positions, and only minor bias in conductance estimates.
> These results suggest that the diagonal approximation is effective in practice for the models and regimes considered in this work. However, we acknowledge its limitations and view it as a first step. The approximation may become insufficient when incorporating more complex channel dynamics—such as calcium-gated conductances—that may introduce stronger intra- and inter-compartment dependencies. A promising direction for future work is to explore more structured covariance approximations, such as block-diagonal (e.g., one block per compartment) or block-diagonal plus low-rank structures, which may better capture the biophysical correlations present in multi-compartment neuron models, while preserving computational efficiency.
>
> - The reviewer suggests applying the method to real Neuropixels or MEA data. We agree, and this is a key direction for future work. Our current study focuses on establishing a principled, robust inference framework in controlled settings with known ground truth. Real recordings introduce additional complexities. We address these in the conclusion and plan to extend our approach to real datasets such as electrical images obtained from retinal ganglion cells.
>
> - The reviewer asks for a deeper discussion of parameter identifiability and whether the identifiable subspace can be characterized for complex models. We thank the reviewer for raising this important point. As discussed in the computational neuroscience literature (e.g., Sterratt et al., 2011), increasing the number of unknown parameters—particularly when morphological properties such as compartment radii or axial resistivity are also unknown—significantly increases the risk of parameter degeneracy. To mitigate this, our experiments make simplifying assumptions: we treat the radii and lengths of compartments as known and fixed. This reduces the effective parameter space and improves identifiability.
> In our smaller simulated models (e.g. 12-compartment neurons), we did not observe signs of degeneracy: across multiple random initializations, the optimizer consistently converged to the same parameter values and marginal likelihood (matching the simulated ground truth), suggesting identifiability.
> For the larger CA1 pyramidal neuron, however, we observe a different behavior. While the marginal likelihood converges reliably to the known optimum, the inferred conductance values continue to drift during optimization. Despite these changes, the membrane voltage remains accurately reconstructed. This suggests the presence of a non-identifiable subspace in which multiple parameter combinations produce indistinguishable outputs.
> Identifying the full identifiable subspace analytically is, to our knowledge, intractable due to the nonlinear and high-dimensional structure of Hodgkin–Huxley-type models. However, several practical strategies can help reduce parameter degeneracy even for larger models like the pyramidal cell. One example, proposed by Migliore and Shepherd (2002), is to constrain conductance densities via spatial parameterizations (e.g., uniform or linearly varying gradients along dendrites). These strategies can reduce the number of free parameters, but must be tailored to the specific cell type.
> We did not implement such parameterizations in this work, as we aimed to develop a general method for recovering conductance parameters that explain observed extracellular voltages in a general setting, without relying on cell-type-specific assumptions. However, these spatial conductance distribution models can be readily incorporated into our method via simple reparameterizations, and we agree they should be included in future work focused on specific neuron types. We will update the manuscript to clarify this point.
>
> - The reviewer recommends evaluating alternative inference approaches. We agree with the reviewer and appreciate the suggestion. Although not included in the paper, we did experiment with the Unscented Kalman Filter (UKF), which showed performance comparable to the EKF when the number of compartments was low. However, UKF is known to scale poorly with latent dimensionality, which limits its applicability to the larger multi-compartment models considered in this work. For these reasons, we focused on EKF (with diagonal approximation) as a scalable and effective inference method.

---

> > ### Comment · Reviewer_ntGU · 2025-08-07
> >
> > Thank you to the authors for their detailed response. It would have been helpful to mention the quantitative results for explorations that they have performed in the past, such as the use of the full EKF and UKF instead of the diagonal approximation to the EKF covariance, in order to fully assess the relative utility of the approximation and comparisons with related methods. Moreover, the addition of Neuropixel or MEA data would have strengthened this paper further, but it remains an interesting approach to an often understudied application. I maintain my positive score.

---

> > > ### Author Response · Authors · 2025-08-09
> > >
> > > Thank you very much for the feedback and for advocating the acceptance of this paper.

---

### Official Review · Reviewer_p6UN · 2025-07-15

**Clarity:** 4
**Significance:** 4
**Originality:** 3
**Rating:** 5
**Confidence:** 2

**Summary:**

This paper introduces a method for fitting multi-compartment Hodgkin–Huxley neuron models using only extracellular recordings from high-density probes. By combining a differentiable simulator (in JAX) with an Extended Kalman Filter (EKF), the method estimates hidden states (e.g., membrane potentials, ion channel dynamics) and learns key biophysical parameters through gradient-based optimization. Validated on synthetic data with both simplified and complex morphologies, the approach recovers conductances, membrane voltages, and neuron position. Fitting full biophysical neuron models from extracellular signals is a novel contribution.

**Questions:**

1. The paper should provide a brief quantitative or visual analysis comparing the full EKF, diagonal EKF, and simpler loss functions (e.g., MSE minimization). This would help readers understand the accuracy vs. efficiency trade-off, and under what conditions the approximations begin to significantly affect parameter recovery.
2. It should emphasize that applying it to in vivo data with overlapping neuronal sources will likely require additional modeling layers or priors (e.g., from anatomy or known cell types).
3. The authors should provide efficient code for easy reproduction. They should consider including benchmarks and documentation on how to run the model for small vs. large neurons, to support adoption and experimentation.

**Ethical Concerns:**

["NO or VERY MINOR ethics concerns only"]

**Limitations:**

Yes.

**Paper Formatting Concerns:**

None.

**Quality:**

3

**Strengths And Weaknesses:**

Strengths:

1. The combination of high-density extracellular recordings with multi-compartment modeling is novel. By casting the neuron and electrode system as a state-space model and using an EKF, the authors can continually update the latent membrane states and use those to optimize parameters.

2. The method should be scalable. The paper avoids black-box training of a neural network, instead it directly optimizes the mechanistic model’s likelihood, which ensures the learned parameters are physically interpretable. Also, the authors implement efficiency improvements, e.g. using a sparse Jacobian computation for the state transition (via the SPARSEJAC library) so that they never form the full dense Jacobian of the ODE system, and adopting a diagonal approximation to the EKF covariance to reduce complexity.

3. The paper demonstrates the approach on realistic neuron morphologies taken from the literature (a 2D retinal ganglion cell and a 3D CA1 pyramidal neuron). The method scaled to over 100 compartments with many parameters. These experiments show that the algorithm can recover key parameters like sodium/potassium conductances and the neuron’s location in space from synthetic extracellular data.

4. The paper is generally well-written and structured. Technical details of the model (Hodgkin–Huxley equations, extracellular potential calculation) are provided, and the state-space formulation is clearly described. The authors also address reproducibility: they use open-source tools (JAXLEY, DYNAMAX) and plan to release code and instructions.

Weaknesses:

1. The approach makes some assumptions that may limit its real-world applicability. First, the neuron’s morphology (branching structure and compartmental layout) is assumed to be known and fixed during fitting. In practice, for an arbitrary neuron recorded extracellularly, the morphological structure would not be directly known. This limitation is understandable, but it should be clearly stated. Similarly, the method currently fits one neuron at a time in isolation, i.e.  it does not account for overlapping signals from multiple neurons.

2. The study is validated only on synthetic data. There are no real-world experiments or noise beyond what the model assumes (Gaussian noise). While this is completely reasonable for a first proof-of-concept, it remains to be seen if an actual extracellular recording from an unknown neuron can be fitted with this approach. However, the authors do discuss the challenges of real data in their conclusion (e.g. electrode drift, unknown ground truth).

---

> ### Author Rebuttal · Authors · 2025-07-31
>
> We thank the reviewer for taking the time to review our submission and for their positive comments! We were particularly pleased by the comments on the novelty and clarity of our work. Below, we address each of the three suggestions made by the reviewer in the "Questions" section:
>
> - The reviewer suggests including a brief analysis—either visual or quantitative—comparing the performance of the full EKF, diagonal EKF, and MSE-based optimization. We thank the reviewer for this helpful suggestion. We agree that such a comparison would strengthen the paper by providing concrete evidence of the relative performance of each method. We will include a quantitative and visual analysis comparing full EKF, diagonal EKF, and MSE minimization in the camera-ready version.
>
> - The reviewer notes that applying our method to in vivo recordings will likely require additional modeling layers. We fully agree and will revise our conclusion to emphasize this point more strongly.
>
> - We fully agree with the reviewer on the necessity of providing efficient, well-documented code—including benchmarks and usage instructions for both small- and large-scale models. This is part of our plan, and it will be made available in a public repository by the camera-ready deadline.

---

### Official Review · Reviewer_co5q · 2025-07-21

**Clarity:** 3
**Significance:** 3
**Originality:** 3
**Rating:** 5
**Confidence:** 3

**Summary:**

This work develops an approach for fitting multi-compartment Hodgkin-Huxley models using dense extracellular voltage recordings. Previous work developed differentiable simulators to fit biophysical neuronal models from intracellular recordings. This work adds a straightforward biophysical model of extracellular potentials, as well as a scalable Extended Kalman Filter (EKF) framework to improve parameter estimation given noise and modeling uncertainty. The approach is tested on several biophysical models of increasing complexity, from a single-compartment model to models of retinal ganglion cells and hippocampal pyramidal cells. Across these scenarios, the method shows good ability to reconstruct channel conductances, voltage time courses, and the geometry of the cell, and markedly outperforms a version of the method without the EKF component.

**Questions:**

- In Sections 4 and 5, when the authors state the dimensionality of the latent space, and the number of parameters that need to be estimated, could they spell out more explicitly how these numbers arise from the model definition and simulation setup?
- In Section 5.1 (and more generally) why is there estimation bias in the high-noise regime?
- Throughout the experiments, could the authors help the uninitiated reader appreciate whether or not, in Fig. 4 for example, a bias of 4e-5 is significant? At what order of magnitude would the bias present significant issues from a modeling perspective?
- In Figure 4 D and E, in the high-noise regime, why are the voltage traces so faithfully reconstructed despite bias in the conductance parameter estimates? Does this point relate to unidentifiability/degeneracy discussed in subsequent sections?

**Ethical Concerns:**

["NO or VERY MINOR ethics concerns only"]

**Final Justification:**

This work introduces an interesting framework for estimating biophysical neuronal models from extracellular voltage recordings. The limitations pointed out by myself and other reviewers are well acknowledged by the authors, and the authors' proposed changes will improve the clarity and framing of the paper. I continue to think this work is a valuable and interesting contribution, and will maintain my score.

**Limitations:**

yes

**Quality:**

3

**Strengths And Weaknesses:**

Strengths:

While the core methods in this paper seem to be a combination of largely off-the-shelf tools, the work is well executed. In general, I found the paper to be quite readable. The modeling and inference choices were reasonable. The evaluations were comprehensive and followed a logical build-up. Modern ultra-dense extracellular probes like Neuropixels are most commonly used for neuronal population recordings, but the use of these probes to estimate detailed biophysical models is an interesting alternative application.

Weaknesses:

The novelty is slightly limited. The differentiable simulation, EKF, and intra- and extracellular biophysical modeling techniques used are all well established. But again, their combination was well executed here. There are also some minor areas where clarity could be improved (see Questions, below).

---

> ### Author Rebuttal · Authors · 2025-07-31
>
> Firstly, we would like to thank the reviewer for taking the time to review our submission and for their positive comments! Below, we address each of the four questions raised and hope our responses provide the necessary clarifications:
>
> - The reviewer asks for a clearer explanation of how the stated latent space dimensionality and the number of parameters being estimated arise from the model and simulation setup. We agree with the reviewer that the dimensions of the latent and parameter spaces should be made clearer, and we will revise the paper accordingly. In general, the latent space at time $t$ includes the membrane voltage in each compartment as well as the full set of gating variables defined for all the ion channels. If the model has $ N $ compartments and compartment $i$ has $n_i$ gating variables, the latent state at time $t$ has dimension $N + \sum_{i=1}^N n_i$. As for the parameters being estimated, we include the maximum conductances of the ion channels and, when specified, additional geometric parameters related to the neuron's spatial position relative to the probe.
>
> - The reviewer asks why parameter estimates become biased when the observation noise is large. When measurement noise is high, the EKF places less weight on the observations and relies more heavily on its internal predictions of the system’s state, which are based on the current parameter estimates. Since these parameters are still being learned, it means that the filter adapts more slowly, and the estimates tend to stay closer to their initial values. Another way to understand this effect is through the marginal likelihood: as measurement noise increases, the gradient of this objective shrinks in magnitude, flattening the landscape. A flatter surface makes it easier for optimization to stall in regions that are close to—but not exactly at—the true parameter values.
>
> - The reviewer asks when bias should be considered significant from a modeling perspective. Bias becomes a concern when it leads to inaccurate reconstruction of the membrane voltage or other latent states. In our setting, the primary goal is to estimate biophysical parameters that enable accurate recovery of the membrane and extracellular voltages. If the inferred parameters achieve this—despite being biased relative to ground truth— while being consistent with typical values found in the literature, we consider the result acceptable. In Fig. 4, for instance, we observe some bias in the estimated conductances, but the reconstructed voltages remain highly accurate, so the bias is not problematic in practice.
>
> - Finally, the reviewer asks why the voltage reconstructions in the high-noise regime remain accurate for the retinal ganglion cell, even when the conductance estimates are biased, and whether this relates to the identifiability issues discussed later. We believe this is an insightful observation and agree that the accurate voltage reconstructions, despite parameter bias, reflect the degeneracy discussed later on. Specifically, multiple configurations of conductances can produce nearly indistinguishable voltage dynamics. We acknowledge that this connection should have been made earlier in the paper, and we will revise the relevant sections to clarify the role of parameter non-identifiability.

---

> > ### Comment · Reviewer_co5q · 2025-08-06
> >
> > I appreciate the authors' responses to my and the other reviewers' comments. The authors' proposed changes will improve the clarity and framing of the paper. I continue to think this work is a valuable and interesting contribution, and will maintain my score.

---

> ### Author Response · Authors · 2025-08-09
>
> Thank you very much for the feedback and for advocating the acceptance of this paper.

---

### Note · Authors · 2025-08-16

We thank Reviewers co5q, p6UN, ntGU, and tGNX for their positive and thoughtful feedback. They highlighted the novelty of combining high-density extracellular recordings with multi-compartment Hodgkin–Huxley modeling, the scalability of our Extended Kalman Filter–based framework, and the clarity of our presentation. Reviewers also appreciated the comprehensive evaluations and the method’s broad applicability across different probes and morphologies. They offered valuable suggestions to further strengthen the work—such as clarifying latent space dimensionality and bias interpretation, releasing well-documented code with benchmarks, and providing a deeper discussion of parameter identifiability—which we will incorporate into the camera-ready version.

Reviewer aSRG expressed a divergent opinion. We addressed their concerns in detail in our rebuttal, but no further engagement or follow-up discussion occurred.

---

### Decision · Program_Chairs · 2025-09-17

**Decision:**

Accept (poster)

**Comment:**

This paper received largely positive reviews (barring one reviewer). The application is a novel and important one, and the approach is well-executed, making this a good applied neuroscience paper, even if some of the ML is a bit incremental/off the shelf. The biggest weakness concerned the absence of experiments with real-world recordings, though the reviewers were largely of the opinion that this is not fatal. Please go over the discussion with the reviewers when preparing the camera-ready version.